# Whole-Exome Sequencing of 24 Spanish Families: Candidate Genes for Non-Syndromic Pediatric Keratoconus

**DOI:** 10.3390/genes14101838

**Published:** 2023-09-22

**Authors:** Carmen González-Atienza, Eloísa Sánchez-Cazorla, Natalia Villoldo-Fernández, Almudena del Hierro, Ana Boto, Marta Guerrero-Carretero, María Nieves-Moreno, Natalia Arruti, Patricia Rodríguez-Solana, Rocío Mena, Carmen Rodríguez-Jiménez, Irene Rosa-Pérez, Juan Carlos Acal, Joana Blasco, Marta Naranjo-Castresana, Beatriz Ruz-Caracuel, Victoria E. F. Montaño, Cristina Ortega Patrón, M. Esther Rubio-Martín, Laura García-Fernández, Emi Rikeros-Orozco, María de Los Ángeles Gómez-Cano, Luna Delgado-Mora, Susana Noval, Elena Vallespín

**Affiliations:** 1Molecular Ophthalmology Section, Medical and Molecular Genetics Institute (INGEMM) IdiPaz, Hospital Universitario La Paz, 28046 Madrid, Spain; carmenglezatienza@gmail.com (C.G.-A.); eloisasancaz@gmail.com (E.S.-C.); prsolana@salud.madrid.org (P.R.-S.); mariarocio.mena@salud.madrid.org (R.M.); crodriguezj@salud.madrid.org (C.R.-J.); victoriaeugeniafdezmontano@hotmail.com (V.E.F.M.); copatron@salud.madrid.org (C.O.P.); m.esther.rubio@salud.madrid.org (M.E.R.-M.); lgarciaf@salud.madrid.org (L.G.-F.); 2Department of Pediatric Ophthalmology, IdiPaz, Hospital Universitario La Paz, 28046 Madrid, Spain; nvilfer@gmail.com (N.V.-F.); almudena.hierro@salud.madrid.org (A.d.H.); anaboto@me.com (A.B.); martu_gue@hotmail.com (M.G.-C.); natalia.arruti@salud.madrid.org (N.A.); irene.rosa@salud.madrid.org (I.R.-P.); juancarlos.acal@salud.madrid.org (J.C.A.); yoana.blasco@yahoo.es (J.B.); marta.naranjo@salud.madrid.org (M.N.-C.); susana.noval@salud.madrid.org (S.N.); 3European Reference Network on Eye Diseases (ERN-EYE), Hospital Universitario La Paz, 28046 Madrid, Spain; 4Biomedical Research Center in the Rare Diseases Network (CIBERER), Carlos II Health Institute (ISCIII), 28029 Madrid, Spain; beatriz.ruz@salud.madrid.org (B.R.-C.); emikarina.rikeros@salud.madrid.org (E.R.-O.); lunadelde@gmail.com (L.D.-M.); 5Clinical Bioinformatics Section, Medical and Molecular Genetics Institute (INGEMM) IdiPaz, CIBERER, Hospital Universitario La Paz, 28046 Madrid, Spain; 6Clinical Genetics Section, Medical and Molecular Genetics Institute (INGEMM) IdiPaz, CIBERER, Hospital Universitario La Paz, 28046 Madrid, Spain; mariadelosangeles.gomez@salud.madrid.org

**Keywords:** non-syndromic pediatric keratoconus, whole-exome sequencing, ophthalmogenetics

## Abstract

Keratoconus is a corneal dystrophy that is one of the main causes of corneal transplantation and for which there is currently no effective treatment for all patients. The presentation of this disease in pediatric age is associated with rapid progression, a worse prognosis and, in 15–20% of cases, the need for corneal transplantation. It is a multifactorial disease with genetic variability, which makes its genetic study difficult. Discovering new therapeutic targets is necessary to improve the quality of life of patients. In this manuscript, we present the results of whole-exome sequencing (WES) of 24 pediatric families diagnosed at the University Hospital La Paz (HULP) in Madrid. The results show an oligogenic inheritance of the disease. Genes involved in the structure, function, cell adhesion, development and repair pathways of the cornea are proposed as candidate genes for the disease. Further studies are needed to confirm the involvement of the candidate genes described in this article in the development of pediatric keratoconus.

## 1. Introduction

Keratoconus (KC) is one of the most significant corneal disorders worldwide and is one of the main causes of corneal transplantation in developed countries. It is characterized by progressive thinning and the formation of a cone-shaped bulge in the cornea that can lead to a significant reduction in total visual acuity and quality of life in patients [1,2].

The pathology was described in 1854 by Nottinghem, and its terminology comes from the Greek words “Keras” (cornea) and “konos” (cone) [2]. Keratoconus is a multifactorial corneal disease, the development to which genetic and environmental factors contribute and which generally presents bilaterally and asymmetrically, with one eye more affected than the other [1]. Generally, it is usually an isolated ocular pathology, but it can appear together with other ophthalmologic diseases (Leber congenital amaurosis (LCA) or Fuchs Endothelial Corneal Dystrophy (FECD)) or diseases of a systemic nature (Ehlers–Danlos syndrome, Marfan syndrome or Down syndrome) [2,3,4,5].

The onset of the disease usually occurs between the second or third decade of the patient’s life and continues its progression until it stabilizes in the fourth decade of life. However, keratoconus can begin at any age, though not very prevalent in patients over 50 years of age, and can occur in childhood with worse progression and more severe symptoms [6,7,8].

### 1.1. Epidemiology and Etiology of Keratoconus

Although the National Institute of Health of the Genetic and Rare Diseases Information Center (GARD) originally classified keratoconus as a rare disease due to its incidence of less than 1/2000 persons, reviews in the scientific literature on the pathology point to a higher occurrence. Furthermore, the present time has been pointed out as a moment of increasing prevalence of the disease [6,9]. The prevalence and incidence ranges of keratoconus are estimated to be between 0.2 and 4.79 per 100,000 people and between 1.5 and 25 per 100,000 people/year, respectively. The variability of these parameters depends on geographic location and ethnicity, with the highest values in Asia and the Middle East, where more than 20 persons per 100,000 persons/year are reported, and in Iran, with an incidence of 22.3 patients per 100,000 persons/year. In the Caucasian population, the estimated incidence is 2–4 persons per 100,000 persons/year [2]. In 2015, in Spain, the estimated prevalence of the pathology was 181 cases per 100,000 inhabitants, being twice as frequent in men than in women [7,10]. The prevalence and incidence values are estimated, as their calculation for the disease is complicated given the clinical and genetic heterogeneity of KC, the presence of mild symptoms in the early stages of the disease and the different criteria used in each study for its classification [1,2].

The clinical symptomatology of keratoconus is variable, although thinning of the corneal stroma in the central and paracentral parts always predominates. KC can lead to significant visual impairments, such as myopia, light sensitivity, visual distortion or irregular astigmatism [2]. In addition, patients may present with clinical signs such as Vogt’s striae (vertical lines appearing in the superficial stromal layers); Fleischer’s ring (a ring that forms at the base of the corneal cone due to ferritin deposition in the epithelium); the appearance of small scars on the cornea that reduce vision; enlarged and more visible corneal nerves; fibrillary subepithelial lines; hydrops (acute inflammation of the stroma due to rupture of Descemet’s membrane); and, in the most advanced cases, Munson’s sign (a wedge-shaped indentation of the lower eyelid when the patient looks down) [8].

### 1.2. Pediatric Keratoconus

When the disease appears in children under 18 years of age, it is called pediatric keratoconus. It is more severe than adult keratoconus and accounts for 15–20% of all corneal transplants in children to stop the progression of the disease. This points out the need for an early diagnosis of the pathology in order to proceed with the therapeutic approach as soon as possible. Some articles point out an inverse correlation between the age of diagnosis of the disease in pediatrics and the progression and severity of the disease. At the time of diagnosis, 30% of pediatric patients present stage IV, while in adult patients, this occurs in 8% [11,12].

In children, KC manifests with a more significant and rapid progression, and symptoms such as blurred vision, polymetropia and defective vision appear. The early onset of KC is usually associated with a family history, ocular allergy, atopy with ocular rubbing and systemic syndromes, and it is especially associated with allergic keratoconjunctivitis, cataracts and glaucoma [11,12].

Although keratoconus is a multifactorial disease, the incipient onset of the disease in pediatric patients, who are less exposed to external factors, could indicate the presence of mutations in genes involved in the pathophysiology of the disease. In this sense, genetic diagnosis could be a useful diagnostic tool in children with keratoconus. Furthermore, the use of genetic markers could prove to be a helpful diagnostic approach due to the great variability in clinical examinations performed in pediatric patients and the difficulty in identifying mild symptoms at the onset of the disease [11].

### 1.3. Diagnosis, Classification and Treatment of Keratoconus

Early detection of keratoconus is key to controlling its progression and reducing the need for corneal transplantation in the most advanced stages. Diagnostic methods are based on corneal morphology using optical coherence tomography (OCT), which studies the corneal thickness profile (reduction in epithelial basal cells and fragmentation of the anterior limiting lamina); the study of tomographic parameters through tomographs, which allows the determination of anomalies of posterior corneal elevations; and the study of corneal biomechanics using corneal pachymetry, topography and tomography, since in the pathophysiology of the disease, structural proteins (collagen) seem to be altered [2,6,12].

There are different ways to classify keratoconus: according to its morphology, disease progression and a series of indices and scores obtained from corneal topography. The classification based on the use of the PENTACAM^®^ tomograph assesses severity and progression based on changes in the corneal volume, angle and depth and provides a topographic classification, adjusted to the classic Amsler or Muckenhirn states. PENTACAM^®^ allows the classification of KC according to the morphology (croissant, duck, snowman, X, nipple and bowtie) and according to the grade (Table 1) [13,14].

The severity of disease presentation determines the type of treatment the patient undergoes. In patients with a genetic predisposition to develop the disease, the avoidance of ocular rubbing, the use of topical antiallergic therapy (in patients with atopy), the use of treatment against allergic keratoconjunctivitis (in required cases) and frequent topographical follow-up are advised as preventive methods. For pediatric patients, in the mildest cases, spectacles are used. When KC progresses, the treatment of choice is visual rehabilitation with rigid gas-permeable contact lenses, multicurve lenses and corneoscleral/scleral lenses [12].

In severe cases, the patient may undergo early collagen cross-linking therapy. This technique uses ultraviolet A (UV-A) light and vitamin B2 or riboflavin as a photosensitizer with the goal of increasing the biomechanical stiffness of the cornea by increasing the cross-linking of proteoglycans and collagen and their proper arrangement [12,15].

The rapid progression of pediatric keratoconus makes it the most common non-traumatic indication for pediatric keratoplasty. The use of intracorneal Ferrara rings (ICRS) in studies has shown visual results in approximately 43% of children. The treatment of choice is deep anterior lamellar keratoplasty (DALK). In more severe cases with ectasia, corneal transplantation is used [12,15].

Although there are different treatments involved in treating the symptomatology of the disease, none of them is completely effective, which shows the need to know the genes involved in the pathophysiology of keratoconus that may serve as possible therapeutic targets.

### 1.4. Structure and Development of the Cornea and Its Histological Changes 

#### 1.4.1. Structure of the Cornea

In order to understand the different histological changes that occur in the disease, it is necessary to know the different structures that constitute the cornea. The cornea is an avascular tissue formed by the corneal epithelium, the epithelial basement membrane, Bowman’s membrane, the stroma and the corneal endothelium [16].

The epithelium consists of non-keratinized, stratified squamous epithelial cells covered with a glycocalyx that maximizes the surface area of the mucous layer of the tear film on the outermost surface of the cornea. Between the cells, there is an important anchoring complex formed by hemidesmosomes and cadherins that represent a link between the intracellular cytoskeleton of the basal epithelial cell and the posterior stroma. The main functions of this layer are to prevent the dehydration of the cornea and to act as a surface barrier for immune regulation and blocking the entry of pathogens [16].

The epithelium is separated from the stroma by an epithelial basement membrane that is deposited by the basal epithelial cells and is divided into a dense and lucid lamina composed of collagens, laminins, proteoglycans (heparan sulfate) and nidogens. Heparan sulfate, perlecan, mediates the migration, survival and differentiation of keratinocytes that maintain the corneal epidermis. The membrane acts as a physical barrier and modulates the effect of factors released by the endothelium on cell differentiation and keratocyte apoptosis [16].

Adjacent to the epithelial basement membrane is the acellular Bowman’s layer made up of collagen fibrils, mainly collagen I, although collagens II, V and XII also form part of the structure. Its function is to protect the subepithelial nerve plexus [16].

The stroma constitutes 90% of the volume of the cornea and is organized into a network of collagen fibers and ground substance with an extracellular matrix (ECM) composed of water, inorganic salts, proteoglycans and glycoproteins. Stromal collagen is mainly types I and V. The proteoglycans in this layer (lumican, keratocan, mimecan and decorin) are involved in forming the correct interfibrillar space by binding to the collagen fibrils. The cell type par excellence of the stroma is the keratocytes, which have the function of maintaining the integrity of the layer, producing collagen, glycosaminoglycans and matrix metalloproteinases. The corneal stroma acts as a structural support for the cornea and is involved in corneal transparency by providing the correct differentiation of the keratocytes, the correct arrangement of the collagen fibril lattice and the correct balance between proangiogenic and antiangiogenic factors that maintain the avascularity of the corneal stroma. In the regulation of this process, there are corneal crystallins, where the enzymes aldehyde dehydrogenase class 3 (ALDH3), endothelial growth factor (VEGF) and membrane metalloproteinase type 1 (MMP-1) stand out. The stroma is also involved in corneal immunity, and immature dendritic cells and precursors of the central cornea, resident bone marrow-derived dendritic cells in the peripheral cornea and macrophages are found in this region. Likewise, the keratocytes themselves, when stimulated by TNF-α and IL-1 released by the epithelium, are activated and produce IL-6 and defensins [16].

#### 1.4.2. Development of the Cornea

The cornea has two embryonic origins. On approximately day 33 of gestation, the corneal epithelium is formed from the overlying ectoderm that forms the primitive lens derived from the invagination of the superficial ectoderm located over the neural optic cup. Relevant in this process is the expression of cytokeratins, which changes from initiation to the maturation of K12 and K3, specific to the tissue. The microRNA miR-450b-5p and the PAX6 gene act in the process [16].

On the other hand, the corneal stroma and endothelial layers are derived from periocular mesenchyme (POM) cells, which are derived, in turn, from cranial neural crest cells (NCCs) originating in the neural tube. Two waves of POM migration occur: the first wave will form the endothelial cell layer through a mesenchymal-to-epithelial transition (MET), and the second wave gives rise to the corneal keratocyte population. As embryonic development proceeds, they differentiate into mature keratocytes that begin to synthesize and secrete mature extracellular matrix components (collagen types I, V and VI or keratan sulfate). During stromal development, the binding of glycosaminoglycans and proteins that fill the spaces between collagen fibers, together with keratan sulfate proteoglycans, occurs. Descemet’s membrane is formed by the deposition of collagen secreted by corneal endothelial cells [16].

#### 1.4.3. Histological Changes in Keratoconus

Histological changes have been detected in all of the layers that form the cornea but are more pronounced in the central zone. The main histological changes occur in the corneal epithelium, the anterior limiting lamina or Bowman’s membrane and the stroma (Figure 1) [2].

Changes in the epithelium are more evident with a more severe disease. The reduction in the thickness of the epithelium does not occur in a similar manner, but rather some areas are thinner than others, resulting in a phenotypic pattern. The epithelium loses cellular uniformity, cells change their morphology, and the anterior limiting lamina is damaged. When the epithelium ruptures, the basal cells of the epithelium move toward the anterior limiting lamina, where they accumulate together with ferritin particles, constituting one of the frequent clinical signs of KC: Fleischer’s ring. Epithelial shrinkage also results in another clinical symptom: increased visibility of corneal nerves in the subbasal corneal nerve plexus [2].

In 7 out of 10 keratoconic eyes, a Z-shaped rupture of the anterior limiting lamina is observed due to the entry of epithelial cells, causing the separation of the collagen bundles and the proliferation of collagenous tissue from the stroma. Thus, in the anterior lamina of keratoconic eyes, there are epithelial cells and stromal keratocytes [2].

As the stroma is the main part of the cornea, it is also the most affected layer in keratoconus. In KC, stromal thickness is reduced, vertical and horizontal collagen lamellae are rearranged, the interfibrillar space between collagen lamellae is reduced, and proteoglycans are increased. The new arrangement of the collagen bundles, through confocal microscopy, is observed as dark and light bands that intercalate, producing the clinical symptom of Vogt’s striae [2]. In the stroma, there is also a reduction in keratocytes and the entry of non-keratocytic cells, such as leukocytes, that participate, through the release of catabolic enzymes such as metalloproteinases, in the degradation of the collagen matrix [6].

Descemet’s membrane is affected in the most severe cases, in which it ruptures and the entry of aqueous material from the stroma is clinically translated into corneal hydrops. If the progression continues, it is manifested in the clinic as Munson’s sign. Changes in the endothelial cell density have only been documented in moderate–severe keratoconus [2].

### 1.5. Pathophysiology of Keratoconus

Currently, the pathophysiology of keratoconus is unknown. It is a multifactorial disorder caused by the involvement of different factors: environmental, metabolic or genetic. Enzymatic processes, inflammation and oxidative stress regulation are relevant in this pathology [12,17].

#### 1.5.1. Environmental Factors

Approximately 53% of patients with keratoconus have allergies or have had hay fever, 15% have asthma, 8.5% have atopic dermatitis and 27% have vernal keratoconjunctivitis [12].

Eye rubbing is considered one of the main risk factors, and recent studies estimate that 50% of patients rub their eyes. The study of eye rubbing is retrospective in nature, so it is complicated to conclude whether it is a cause of keratoconus or a result of atopy or allergy [6,17]. In cases of allergy and keratoconjunctivitis, it appears that allergen exposure and its interaction with specific Immunoglobulin E (IgE) in mast cells and basophils leads to the release of vasoactive mediators, such as histamine, and eye rubbing results in microtrauma to the corneal epithelium, leading to an increase in inflammatory mediators (IL-6 and TNF-α) and membrane metalloproteinases (MMP-1, MMP-2, MPP-9 and MMP-13) in epithelial and stromal cells. The release of inflammatory mediators promotes the release of IL-1, which, together with MMP-13, participates in keratocyte apoptosis and thus in the reduction in stromal volume as a consequence of extracellular matrix degradation, resulting in corneal thinning [12].

#### 1.5.2. Role of the Inflammatory Process

Keratoconus has been commonly described as a “Non-inflammatory” disorder. However, tears from keratoconic eyes show elevated levels of inflammatory mediators (IL-1β, IL-6, TNF-β, TNF-γ) and reduced levels of anti-inflammatory mediators (IL-10), which could be evidence of the importance of an inflammatory role in the pathology [12,18].

#### 1.5.3. Role of Enzymes

There are numerous articles reporting elevated levels of metalloproteinases in keratoconus. MMP-1, MMP-2, MMP-7, MMP-9 and MMP-13 stand out, which participate in the degradation of fibronectin, membrane glycoproteins and collagens I and II, of vital relevance in the adequate deposition of the ECM and in the apoptosis of keratocytes [12,19]. The enzyme lysyl oxidase (LOX) oxidizes lysine and hydroxylysine residues in collagen and allows the formation of covalent cross-links between collagen fibers and elastin. Reduced LOX in KC is associated with the loss of cohesion between collagen fibrils and corneal ectasia [12,20].

#### 1.5.4. Gender

Keratoconus is a pathology that affects both genders. There are epidemiological studies that point to a prevalence 2 to 5 times higher in men and others that point to a higher prevalence in women. Several studies report changes in sex hormones in keratoconus. A case has been reported of a 49-year-old woman with frustre keratoconus who, after treatment for endometriosis with an estrogen activity regulator with tibolone, progressed to keratoconus with the most severe stage. A study carried out in Asturias indicates a higher incidence in men than in women. Another study reports elevated levels of estradiol in the plasma of men with keratoconus with respect to the control population, which could be related to the increase in proinflammatory cytokines (IL-6, IL-1β, IL-8 and GM-CSF) and in the activity of MMP-2 and MMP-9 involved in the etiology of keratoconus. Furthermore, the tendency to develop KC is slightly earlier in women [10,17,21,22].

#### 1.5.5. Genetic Factors

Keratoconus is a disease with great genetic heterogeneity. Inheritance of the pathology can be autosomal recessive, autosomal dominant (with incomplete penetrance) or sporadic. It is estimated that approximately 5–20% of patients with keratoconus have a positive family history. The prevalence of keratoconus in first-degree relatives is approximately 20.5%, 15–67% higher than in the general population [6,12].

The association of keratoconus with different genetic syndromes, such as Down syndrome, Marfan syndrome, Ehlers–Danlos syndrome, osteogenesis imperfecta or Leber congenital amaurosis, emphasizes the importance of the genetic background in the pathology [12].

Although the genetic study of the disease is complex, given the great heterogeneity and incomplete penetrance of the disease, genetic variants have been detected in patients with the disease in candidate genes possibly involved in the pathology. Candidate genes include genes described for other corneal dystrophies; genes with antioxidant function; genes involved in the structure of the cornea, in the synthesis and proper deposition of collagen in the extracellular matrix and in the interaction between cells and collagen proteins; and genes that regulate the mechanical properties of the corneal connective tissue [1].

Genes described in corneal dystrophies and other ocular diseases.

Variants have been found in patients with keratoconus in genes associated with corneal dystrophies. Genes that stand out include *VSX1* for its role in posterior polymorphous dystrophy (PPCD); *ZNF469*, implicated in brittle cornea syndrome and central corneal thinning (CCT); and *DOCK9* or Zizimin1, associated with corneal ectasia [1,23].

Genes involved in corneal development.

The *VSX1* gene encodes for the visual system homeobox 1 protein containing CVC domains and *homeodomains* that function as repressors by binding to the nucleus in a region of cone opsin genes in early stages of development controlling the differentiation of retinal bipolar cells. Mutations in this gene can prevent the binding of the homeobox protein to DNA and the correct development of the retina [1,24].

Genes involved in corneal structure.

Variants in genes involved in the maintenance of the composition of the different layers that compose the cornea have been related to keratoconus. All corneal layers present a common and predominant structural component, collagen, and thus, variants in genes involved in collagen synthesis have been described that result in a decrease in stromal collagen, mainly type I, II and IV collagen. Variants have been detected in *COL4A3* and *COL4A4* genes [1,6,25]. In addition, as 10–20% of total corneal collagen is collagen V, mutations in the genes coding for its α1 and α2 chains, *COL5A1* and *COL5A2*, respectively, have been associated with corneal thinning and keratoconus [12,26]. Likewise, variants have also been described in genes involved in the proteolytic processing of collagen, such as BMP1, responsible for the proteolysis of pro-COL5A1 [1]. *ZNF469* has been described as a candidate gene for its function as a transcription factor in the synthesis and organization of collagen fibers [1].

The correct deposition of collagen and the rest of the components of the extracellular matrix is essential in the maintenance of the transparency and functions of the cornea; studies have related gene variants in *LOX* with the development of the disease due to its role in the cross-linking of collagen and elastin in the extracellular matrix [1,6,20].

For the maintenance of the structural integrity and transparency of the cornea, the proper attachment of the cell types that compose its layers is necessary. Cell adhesion is regulated by the *DOCK9* gene, which translates into a guanine nucleotide exchange factor (GEF) that activates CDC42 [1]. Although its mechanism of action is still unknown, mutations in *DOCK9* have been detected in families with this pathology [6,27].

Genes involved in corneal function.

Mutations in the *MIR184* gene have been identified in the development of KC. MicroRNA 184 (the most abundant miRNA in the corneal epithelium) interacts with miRNA miR-205 in corneal repair by regulating VEGF and Akt signaling pathways, inhibiting corneal angiogenesis [1].

Variants in the *SOD1* gene found in patients impede its antioxidant function, producing an increase in ROS that modify the mitochondrial membrane potential and lead to altered proteins, enzymatic degradation, cellular dysfunction and DNA damage, resulting in increased apoptosis of corneal fibroblasts and, therefore, in keratoconus [12,28].

Given the multifactorial character of keratoconus, whose etiology involves genetic and environmental factors, it is suggested that there is a correlation between the genetic background of pediatric patients and the keratoconus phenotype. It is also speculated that keratoconus is an oligogenic disease, meaning that its development is not due to a single mutated gene but to a cluster of genes.

There are few articles on the genetic study of keratoconus; however, the increased prevalence of the disease in patients with a positive family history suggests the importance of carrying out genetic studies.

## 2. Materials and Methods

The clinical ophthalmologic and genetic approaches to patients with keratoconus were performed in the Ophthalmology Unit and the Ophthalmogenetics Unit of the Institute of Medical and Molecular Genetics (INGEMM) of the Hospital Universitario La Paz de Madrid (HULP), respectively, in accordance with the principles of the Declaration of Helsinki and were approved by the ethics committee.

For patient selection, 24 unrelated families of probands diagnosed with keratoconus during childhood or adolescence were recruited at the Hospital Universitario La Paz. The study consisted of 24 probands and 1–2 of the parents in 20 families (63 participants in total). No families are related by consanguinity.

Inclusion criteria were (1) patients with keratoconus who were ≤18 years at the age of diagnosis and seen at HULP; (2) patients meeting the topographic criteria for KC; (3) mature minors or their legal guardians have signed consent to participate in the study; (4) the absence of other ocular diseases (congenital glaucoma, interventional pediatric cataracts, etc.) that could lead to secondary keratoconus; (5) the absence of syndromic phenotypes.

### 2.1. Ophthalmological Studies

The study considered the following items: gender, age at diagnosis, ethnicity, unilaterality or bilaterality of KC, family history of KC, consanguinity, history of other ocular diseases (especially those that have shown a greater association with KC, such as glaucoma, Retinitis Pigmentosa (RP), Leber congenital amaurosis (LCA), retinopathy of prematurity, cataract, Fuchs endothelial dystrophy, vernal keratoconjunctivitis (VKC)) and other systemic syndromes or diseases (especially those most closely related to KC, such as Down syndrome, Ehlers–Danlos syndrome, Marfan syndrome, osteogenesis imperfecta, mitral valve prolapse), atopy (allergy, asthma, dermatitis, allergic keratoconjunctivitis), ocular rubbing, hydrops and corneal transplantation.

The clinical ophthalmologic evaluation of patients and first-degree relatives who signed the consent form was performed by the Ophthalmology Unit of Hospital La Paz.

Clinical interpretation was based on corneal topography with PENTACAM^®^ based on Scheimpflug images in all patients. The topographic parameters recorded were maximum keratometry (Kmax), mean keratometry (Kmed), pachymetry of the thinnest point (TP), topographic astigmatism (TA), corneal power in the least curved meridian in the central 3 mm zone (K1), corneal power in the most curved meridian in the central 3 mm zone (K2) and the asphericity index (Q). For the study of the degree of haze in the meridian corresponding to Kmax, densitometry (DNS) by Scheimpflug imaging was used. For the classification of keratoconus according to its morphological pattern (croissant, duck, snowman, X, nipple and bowtie), the classification of Fernández-Vega [14] is followed, and the classification according to its degree is based on the topographic classification of keratoconus (TKC) [13].

### 2.2. Genetic Studies

For the genetic analysis, genomic DNA extraction from leukocytes in peripheral venous blood samples was performed in the Preanalytical Area of INGEMM using “Che-magic Magnetic Separation Module I” (Chemagen, PerkinElmer, Waltham, MA, USA). The measurement of DNA concentration was carried out through spectrofluorometer quantification using the NanoDrop ND-1000 spectrophotometer (ThermoFisher Scientific, Waltham, MA, USA). Nextera DNA Exome (Illumina DNA Prep with Enrichment) and IDT for Illumina DNA/RNA UD Indexes Sets A, B, C or D, Tagmentation, were used for library preparation. Sequencing was performed on the high-quality sequencer NovaSeq 6000 System of Illumina (San Diego, CA, USA), capturing 19,433 genes using xGenTM Exome Research Panel v2 IDT.

The first bioinformatics analysis was performed by the INGEMM Bioinformatics team through the development of an analytical algorithm that allows the identification of point polymorphisms (SNPs) and insertions and deletions of small DNA fragments (in-dels) in the captured regions of the exome. This analysis requires sample pre-processing, the alignment of reads with a reference genome (Genome GRCh37-hg19) and the identification of variants with respect to the reference genome, as well as their functional annotation and filtering. All stages of the process include parameters that allow the monitoring of the process and quality controls that guarantee the issuance of a reliable report on the variants identified. The software tools used for bioinformatics analysis were trimmomatic-0.36, bowtie2-align version 2.0.6, picard-tools 1.141, samtools version 1.3.1, bedtools v2.26 and Ge-nomeAnalysisTK version 3.3-0. The databases used were dbNSFP version 3.5, dbSNP v151, ClinVar date 20180930, ExAC-1, SIFT ensembl 66, Polyphen-2 v2.2.2, MutationAssessor, release 3, FATHMM v2.3, CADD v1.4 and dbscSNV1.1.

For the study of the genotype–phenotype correlation, a second analysis was performed to evaluate the clinical pathogenic significance of the variants detected in the patients (Figure 2). For this purpose, the first step was the filtering of all variants resulting from the first bioinformatics analysis. An algorithm was developed based on quality (QUAL > 300), allele frequency (Number_sample_run < 6, allele frequency < 0.015) and pathogenicity (variants classified as benign/likely benign by ClinVar and Combined Annotation Dependent Depletion (CADD) databases are discarded). The pathogenicity of the resulting variants after filtering was then studied using the Franklin Database [29]. This database follows the American College of Medical Genetics and Genomics (ACMG) 2015 guidelines [30] for the classification of variants into benign (B), probably benign (LB), probably pathogenic (LP), pathogenic (P) and of uncertain significance (VUS). Once the pathogenicity of the variants had been defined, the possible relationship of those classified as VUS, LP and P with the pathophysiology of keratoconus was studied. For this purpose, we resorted to the use of the following databases: Pubmed [31], GeneCards [32], OMIN [33], ClinVar [34] and GnomAD [35]. All references to each of the genes with their codes (OMIM: #) can be found on the OMIM website [33]. The variants that appeared to affect the correct splicing were checked with the Alamut program. Similarly, for variants that appeared to produce changes in protein structure, we looked at which region of the structure was affected using Uniprot [36] and analyzed the SIFT-score value. The SIFT score predicts whether the sequence change affects protein function and can take values from 0 to 1, where the more deleterious the change, the closer the value is to 0. Those gene variants that could, according to the existing literature, be related to the disease are included in GeneMatcher [37], a platform that allows contact with other scientists who relate gene variants not previously described in the scientific literature with different phenotypes. The variants included in the study were checked with the IGV (Integrative Genomics Viewer). The oligogenic nature associated with keratoconus, the identification of a large number of variants of uncertain significance, and the limited literature on the genetics of the disease make it difficult to select variants in genes that are likely to cause the disease from the exome data. Therefore, the selection criteria for variants in genes that could be candidates involved in the development of the disease are those found in genes previously related to the disease, are present in more than one affected family, meet segregation criteria and/or, due to the function of the proteins for which they code, may be involved in the pathophysiology of the disease.

## 3. Results

### 3.1. Ophthalmological Examination

Twenty-four families were ophthalmologically examined (Table 2). None of the families present consanguinity or are related to each other. Topographic and clinical data are included in Appendix A (Table A1 and Table A2).

A total of 70.8% of the patients were of Spanish origin, 37.5% were from South America (Colombia, Honduras, Ecuador, Dominican Republic or Venezuela), 4.2% were British and 4,2% were from Portugal. Of the 24 probands, 75% are male (*n* = 18) and 25% are female (*n* = 6). 

Of the 24 probands 25% have first-degree relatives with suspected pathological keratoconus (*n* = 6), and another 25% of the probands have first-degree relatives diagnosed with the disease (*n* = 6). The genetic trees of the 24 families with pediatric probands affected by keratoconus are graphically represented in Appendix A (Figure A1). Topographic evaluation of the progenitors allowed us to discover a family history in 20.9% of patients in whom it had been unknown (*n* = 9) and to increase the number of family members with KC by 7% (*n* = 3) in patients with a known history. 

At the time of diagnosis, 83.7% of patients (including frustre and suspected KC) had a bilateral phenotype.

At the time of diagnosis, 86% of the eyes were pathologic, 6.3% were suspicious, 4.2% were frustres, and 6.3% were normal eyes. The following is the proportion of KC grades in the eyes of our patients: grade 1 in 29.2%, grade 2 in 39.6%, grade 3 in 31.3%, grade 4 in 8.3% and grade 0 (normal eyes) in 6.3%. 

In decreasing order of prevalence, we found the following KC patterns: duck (39.6%) and snowman (22.9%) were the most abundant in the sample, followed by croissant (10.4%) and nipple (8.3%), and finally bowtie (4.2%). The remaining 14.7% consisted of normal eyes and eyes with frustre patterns; the latter is not classifiable according to the Fernandez-Vega A. classification of KC because it only presents a posterior elevation alteration; however, this classification requires the presence of an anterior elevation alteration. 

The following data are the means ± standard deviations of the topographic parameters collected: Q of 0.87 D ± 0.51 (minimum −2.65, maximum −0.02), K1 of 45.71 D ± 12.67 (minimum 38, maximum 58), K2 of 50 D ± 17.55 (minimum 44, maximum 66), Kmed of 49.45 D ± 16, 77 (minimum 42, maximum 61), Kmax of 53.4 D ± 17.38 (minimum 46, maximum 67), TP of 493 μm ± 43.77 (minimum 387, maximum 575) and AT of 4.31 D ± 1.89 (minimum 1, maximum 10).

Regarding risk factors, 62.5% of patients had atopy, and 62.5% of patients reported having a rubbing habit. In addition, other abundant antecedents in the samples and with a possible association with KC were obesity–overweight, which was found in 44.2% of the sample, and Meibomian gland dysfunction (MGD), present in 25.2% of these patients.

However, although the following antecedents are well described in the literature as being associated with KC, they were not highly represented in the sample. Anterior stromal corneal dystrophy, retinopathy of prematurity, congenital astigmatism, congenital cataract and PPD were found in only 4.2% (*n* = 1). None of the patients had a history of vernal keratoconjunctivitis (VKC), glaucoma, Retinitis Pigmentosa (RP), Leber congenital amaurosis (LCA), Fuchs’ dystrophy, Ehlers–Danlos syndrome, Down syndrome, Marfan syndrome, osteogenesis imperfecta or mitral valve prolapse.

### 3.2. Molecular Genetics

The 24 families of probands diagnosed at ages 8–18 years, 18 males and 6 females, with 50% having a family history (first-degree relatives with suspected/pathological keratoconus), were studied by performing a genetic analysis of NGS exomes. In 16 families, the genetic study of both parents and the proband was carried out, which allows us to know the segregation of the genetic variants.

From the 19,433 genes of the exome NGS analysis, gene variants were selected following the selection criteria included in Materials and Methods. Most of the variants selected are of uncertain significance (VUS) (53%) (Figure 3). Probably pathogenic (LP) variants represent 5% of the variants found (Figure 4).

Variants are detected in genes described in associated diseases and genes involved in development, structure, cell adhesion, the maintenance of functions, the regulation of oxidative stress and corneal repair. In the present study, most candidate genes are involved in corneal structure. The table lists the gene variants found by family that could be the cause of keratoconus (Table 3).

Most of the variants are not described in GnomAD; the only identified variants previously described were detected in the OFT-00242 family in the *ZNF469* gene, not associated with any phenotype, and in the OFT-00852 family in the *SLC4A11* gene, previously detected in patients with corneal dystrophies.

## 4. Discussion

### 4.1. Ophthalmological Studies

Clinically, the most represented grade of keratoconus in the population studied was grade 3, and the most common morphological pattern of keratoconus was the duck phenotype. This last result differs from that expected according to the study by Fernández-Vega, who suggests that the croissant pattern is the most prevalent. 

The phenotype of keratoconus was bilateral at diagnosis in 83.7% of patients (including frustre and suspected KC). 

In the population studied, the most represented risk factors were atopy and eye rubbing.

It is interesting to add that, thanks to the topographic review of the progenitors, it was possible to discover a family history in 20.9% of the patients in whom it had been unknown (*n* = 9) and to increase the number of relatives with KC in patients with a known history by 7% (*n* = 3). This could suggest the importance of an ophthalmologic study of the parents of pediatric patients with keratoconus.

### 4.2. Genetic Study

Following the procedure described in the methodology, variants were found in genes that, based on the existing literature and/or on the role played by the proteins for which they code, could be involved in the development of keratoconus. Although the search has been exhaustive, the fact that the disease appears to be oligogenic and of variable penetrance makes the selection of variants in candidate genes complex. In addition, the limitations of the Second-Generation Sequencing of the exome, where some regions have low-quality and low-depth reads, may lead to the loss of variants in genes possibly involved in the etiology. Mutations in selected possible disease-causing genes per family are presented below. However, in many families, additional studies (segregation, functional models, etc.) will be necessary for the confirmation of their implications in the development of the disease and for the genetic diagnosis of the patient.

Family OFT-00242.

Male of Spanish origin diagnosed with bilateral topographic KC at age 15. The patient showed atopy (asthma), an eye-rubbing habit, blepharitis and a history of being overweight. He lacked a positive family history of KC. Before any intervention, his right eye had stage 1 KC and a snowman pattern, and his left eye had stage 1–2 KC and a croissant pattern; the TA was 2.4 D and 1.8 D, respectively.

The proband presents a missense variant (p.(Ala3725Thr)) in *ZNF469* in heterozygosis. *ZNF469* (OMIM:612078) in homozygosis is associated with brittle cornea syndrome type 1 (MIM:229200); however, heterozygous mutations in this gene have been found to cause keratoconus with autosomal dominant inheritance [1,38,39]. The variant has been described in GnomAD in a single allele not associated with the phenotype and has been classified as probably pathogenic. It would be interesting to perform a segregation study to know whether the variant is de novo and probably causative of KC.

In the patient, in addition, another variant was detected in a gene proposed, according to the existing literature, as a possible cause of the disease. A trio-exome NGS study suggests *NDRG1* as a candidate gene in the pathogenesis of keratoconus due to its involvement in the organization of the cytoskeleton. A variant affecting the donor site (c.699-2A>G) in heterozygosis was found in the patient and is classified, according to ACMG criteria, as probably pathogenic. Using Alamut, it was found that the variant affects the canonical splicing site and generates a new donor site that could compromise the structure and function of the protein. *NDRG1* (OMIM: #605262) is also an important regulator of the VEFG factor that is essential for the maintenance of the balance between anti-angiogenic and angiogenic factors to maintain corneal avascularity and transparency, relevant in the development of keratoconus [40].

Also, the proband presents a missense variant (p.(Arg615Trp)) in heterozygosis in the *LOXHD1* gene. Although the *LOXHD1* gene (OMIM: #613072) is associated with autosomal recessive deafness, there is a clear association of this gene with corneal dystrophies such as FECD with autosomal dominant inheritance, which may be relevant in the genetic background of keratoconus [32,33].

Family OFT-00273.

Male of Spanish and English origin diagnosed with bilateral topographic KC at age 13. He lacked a known family history of KC and an eye-rubbing habit, and he did not suffer from atopy. Before any intervention, his right eye had stage 2–3 KC, his left eye had stage 2 KC, both eyes had a snowman pattern, and the TA was 10.8 D in his right eye and 3.4 D in his left eye.

In the genetic study of the patient, two heterozygous missense variants were found in the *MMP1* and *MMP2* genes, which code for metalloproteinases 1 and 2, whose relevance in the pathophysiology of KC is known due to their participation in the remodeling of the extracellular matrix [8,12]. Variants in genes encoding for metalloproteinases have been detected in patients with KC [19]. In GnomAD, only the *MMP1* variant is described in four alleles not associated with the phenotype. The base change that occurs in *MMP1* is associated with a SIFT-score value of 0.001, so it seems that the change that occurs generates an inadequate folding of the protein. Using Uniprot, the variant was found to be associated with a defective protein conformation, as it affects the molecular processing of the 100–469 chain of interstitial collagenase. Given the importance of the function of metalloproteinases and, in particular, the participation of MMP-1 in the proper deposition of ECM in the stroma, it can be speculated that the incorrect function of MMP-1 and MMP-2 proteins could lead to ECM disruption and the incorrect cross-linking of collagen bundles, which can generate the Vogt’s striae that appear from grade 2 keratoconus onwards.

The disruption of the proper ECM structure could also be caused by the *COL8A2* gene, where the patient has a missense variant (p.(Glu189Asp)) in heterozygosis not described in GnomAD. Although the *COL8A2* gene (OMIM: #120252) is associated with the phenotypes of FECD type 1 (MIN: #136800) and PPCD type 2 (MIM: #609140) with an autosomal dominant inheritance pattern, *COL8A2* is associated with corneal thinning and keratoconus. In addition, there is a paper where a mutation in that gene (p.(Gln455Lys)) was found in a Caucasian family with Early-Onset FECD and in a relative with KC. Due to the incomplete penetrance of the gene and the importance of collagen in maintaining the complex structure of the corneal ECM, it is proposed that *COL8A2* contributes to the development of the disease [41]. The patient also has another missense variant in heterozygosis in another gene involved in collagen synthesis, *COL4A3*. *COL4A3* (OMIM: #120070) codes for the α3 chain of collagen IV involved in corneal structure. Although this gene is associated with Autosomal Recessive Alport Syndrome (MIM: #203780), variants in it have been detected in patients with keratoconus [1,6,42].

Family OFT-00290.

Male of Spanish origin diagnosed with bilateral topographic KC at age 14. This patient had an eye-rubbing habit, an anterior stromal corneal dystrophy and a history of obesity. At the beginning of the study, he did not report a positive family history or atopy (although his paternal grandmother does suffer from a corneal dystrophy, which is the cause of her blindness). Before any intervention, his right eye had stage 1–2 KC and a croissant pattern, whilst his left eye had stage 3–4 KC and a snowman pattern; the TA was 2.1 D and 4 D, respectively.

The genetic study of the patient reveals a variant of maternal inheritance in heterozygosis in the *PIKFYVE* gene classified as VUS that seems to affect the donor site of splicing. The variant is not described in GnomAD. Using Alamut, the variant was found to affect protein splicing. PIKFYVE (OMIM: #609419) belongs to the family of lipid kinases that regulate the phosphorylation of intracellular phosphatidylinositol (PtdIns). Specifically, the gene encodes for the enzyme PIKFYVE or phosphatidylinositol-3-phosphate 5-kinase type III, which phosphorylates PtdIns and phosphatidylinositol-3-phosphate (PtdIns3P) to generate PtdIns5P and PtdIns(3,5)bisphosphate. The PIKFYVE enzyme is also responsible for regulating endomembrane homeostasis and maintaining the biogenesis of endosome-transporting vesicles from early endosomes. Mutations in *PIKFYVE* are associated with Fleck Corneal Dystrophy (FCD) (MIM: #121850) with autosomal dominant inheritance. FCD is characterized by the presence of small white speckles in the corneal stroma, which, histologically, are distended keratocytes with intracytoplasmic vacuoles filled with lipids and mucoliposaccharides due to the impossibility of adequate endosomal transport in the cells. The fact that the most affected corneal structure in FCD is the stroma and that this, in turn, is the most affected corneal layer in the pathology of keratoconus may point to the possibility of a relationship between the two corneal dystrophies. This relationship was evidenced in a study of a family in which there were affected relatives with both phenotypes [43]. A variant in this gene has also been detected in a Chinese family suffering from keratoconus with autosomal dominant inheritance [44]. Suspecting that the variant in *PIKFYVE* could generate Fleck Corneal Dystrophy, the patient was scheduled for an ophthalmologic examination, where the phenotype was confirmed. A subtle clinical picture was also observed in the mother. The cosegregation of the variant caused it to be re-classified, according to ACMG criteria, to LP. 

In the patient, a missense variant p.(Pro548Thr) in heterozygosis was also found in the *LOXL2* gene. The SIFT-score value of 0.002 predicts that the amino acid change affects protein function, which was tested using Uniprot, where, although the variant does not appear to be described, the amino acid is observed to be in the first position of the 548–751 region of the protein. LOXL2 (OMIM: #606663) belongs to the family of lysyl oxidases involved in the cross-linking of collagen and elastin. Given its role in the maintenance of the corneal ECM, it is proposed as a candidate disease-causing gene [1,45].

Although *LOXL2* has previously been linked to keratoconus and may enhance the development of the disease, the clinical and genetic associations seem to suggest that it is especially the *PIKFYVE* gene that contributes to the development of the disease.

Family OFT-00705.

Male of Spanish origin diagnosed with bilateral topographic KC at age 13. This patient had a positive family history of KC (his father had bilateral topographic KC), atopy (atopic dermatitis, multiple allergies), an eye-rubbing habit and blepharitis. Before any intervention, his right eye had stage 1–2 KC, his left eye had stage 2 KC, the TA was 2.9 D and 5.1 D, respectively, and both eyes presented a duck pattern.

Of the variants found in the genetic study, the missense variant p.(Asp29Ala)) in heterozygosis in the *PPIP5K2* gene stands out. *PPIP5K2* (OMIM: #611648) encodes for an enzyme with kinase and phosphatase activities that mediates the phosphorylation and dephosphorylation of inositol pyrophosphates (IPs) (5-IP7 and IP-8). *PPIP5K2* is associated with autosomal recessive deafness, but there is a paper that proposes it as the cause of keratoconus in an unrelated American family and in an unrelated European family, where two autosomal dominant missense variants were detected in the phosphatase domain of the p.(Ser419Ala) and p.(Asn843Ser) genes, respectively. The gene appears to have incomplete penetrance, as the first variant appears in a family member not affected by KC. To test the relationship between the first variant in *PPIP5K2* and the etiology of keratoconus, an animal model was used, where there is a decrease in the phosphatase activity and hyperkinase activity of the PPIP5K2 protein, which is associated with abnormal corneal thinning and epithelial histological changes [46]. The variant (described in GnomAD on a single allele, not associated with the phenotype and classified as VUS according to the ACMG) affects enzyme activity, according to Uniprot. The fact that the variant cosegregates in the affected subjects (paternal inheritance) and that variants in the *PPIP5K2* gene have been described in keratoconus leads us to propose it as a candidate gene.

In addition, a missense variant p.(Val19Ala) in heterozygosis was found in the *SLC8A3* gene in both affected subjects. The variant appears in GnomAD, where it is described as LP and not associated with the phenotype. The *SLC8A3* gene encodes for a plasma membrane sodium–potassium exchanger that regulates Ca^2+^ homeostasis through Ca^2+^ efflux into the cytoplasm and Na+ influx into the cellular interior. *SLC8A3* is associated with Schnyder’s corneal dystrophy (MIM: #121800), which is a rare autosomal dominant disease characterized by the abnormal deposition of collagen and phospholipids in the corneal stroma that is associated with decreased visual acuity [32]. The presence of mutations in *SLC8A3* appears to lead to an affected stroma; given the importance of the involvement of this corneal layer in the etiology studied, this gene is speculated as a possible causal gene for keratoconus.

Family OFT-00813.

Male of Dominican and Venezuelan origin diagnosed with topographic KC in his right eye and forme fruste KC in his left eye at age 10. This patient presented a positive family history of KC (his father had KC suspect in his right eye), atopy (allergic conjunctivitis), an eye-rubbing habit and a history of obesity. Before any intervention, his right eye had stage 1 KC and a duck pattern, his left eye had forme fruste KC, and the TA was 4.8 D and 3.6 D, respectively.

The genetic study highlights the presence of a missense variant p.(Thr3116Ile) in the *ZNF469* gene in heterozygosis in both patients (father and son). A case has been reported where a male with a homozygous variant in the *ZNF469* gene had brittle cornea syndrome, and his sons with the same variant in heterozygosis exhibited topographic indices compatible with keratoconus [47]. The fact that the gene is mutated in the OFT-00242 family and the extensive literature on the correlation between variants in *ZNF469* and the phenotype of keratoconus seem to suggest that the gene is involved in its development [1,48].

The *TNXB* gene also has a missense variant p.(Arg1337His) in heterozygosis, segregating in all relatives affected. *TNXB* (OMIM: #600985) codes for an extracellular matrix glycoprotein, tenascin XB, and is associated, when mutated, with Ehlers–Danlos syndrome (MIM: #606408). A 2021 study was based on demonstrating the involvement of genetic variants in genes, including *TNXB*, that cause EDS in the etiology of KC. Thus, it is speculated that the gene could contribute to the genetic background involved in the development of the disease [38]. 

A third missense variant p.(Arg675Trp) in heterozygosis appears in the *HSPG2* gene (OMIM: #142461), which encodes for the proteoglycan perlecan. Perlecan is a major component of the corneal basement membrane and an important constituent of the corneal stromal ECM, which helps maintain the proper stratification of the epithelium and provides a physical barrier to the ECM of the eye. Three families with KC present variants in this gene, associated with an abnormal distribution in the stromal ECM [49]. The variant, of maternal origin, could contribute to a worse prognosis and progression of the disease in the proband compared to his father.

Family OFT-00814.

Male of Spanish origin diagnosed with bilateral topographic KC at age 16. This patient presented blepharitis; however, he did not report a positive family history of KC or atopy (although he reported a family history of blindness: maternal great-grandfather and his great-grandfather’s sister), and whether he had an eye-rubbing habit or not is unknown. Before any intervention, his right eye had stage 3 KC, his left eye had stage 2 KC, the TA was 4.7 D and 6.1 D, respectively, and both eyes presented a duck pattern.

Among the variants found, the missense variant p.(Ala247Glu) in heterozygosis in the *LOXHD1* gene, not described in GnomAD, stands out. The possible role in KC is exposed in the OFT-00242 family. A relevant missense variant p.(Glu30508Asp) in heterozygosis is also detected in *TTN*. The variant is described in one allele in GnomAD as probably pathogenic but not associated with the phenotype. Recently, the literature has raised the possibility of a link between *TTN* and the etiology of keratoconus by inducing corneal thinning. In studies performed using trio-exome NGS, variants in this gene appeared in approximately 25% of those affected, thus suggesting that mutated *TTN* is a risk factor in the disease [40]. The *TTN* gene codes for the largest protein in the human body, titin, which is involved in regulating the organization of the cytoskeleton in cardiomyocytes. The protein also interacts directly with the CAPN3 protein, which is involved in the organization of the extracellular matrix. Therefore, *TTN* is suggested as a possible candidate gene due to its indirect action in the organization of the ECM [50].

Family OFT-00815.

Male of Spanish origin diagnosed with bilateral topographic KC at age 14. This patient presented a positive family history of KC (his mother had bilateral KC suspect), atopy (rhinitis and asthma, atopic dermatitis), an eye-rubbing habit and a history of obesity. Before any intervention, his right eye had stage 3 KC and a nipple pattern, and his left eye had stage 1 KC and a bowtie pattern; the TA was 3 D in his right eye and 2.1 D in his left eye.

A frameshift variant in heterozygosis p.(Ala2956fs) with paternal inheritance in the *COL6A3* gene stands out in the proband. The variant is not described in GnomAD and is classified according to ACMG criteria as LP. Mutations in *COL6A3* appear in keratoconus patients, which seems to point to it as a possible candidate gene. The patient also has a missense variant in the same gene with maternal inheritance. Thus, this compound heterozygosity could be compromising the function of collagen VI in the corneal layers and favoring disease progression [51]. Likewise, other missense variants in heterozygosis in genes coding for collagen II and collagen IV, *COL2A1* and *COL4A4*, respectively, were also found in both patients (mother and son). Variants in these genes are related to the risk of suffering keratoconus [1,38]. A fifth missense variant p.(Leu21481Pro) appears in *TTN* with maternal inheritance, and its contribution to KC is exposed in the OFT-00814 family [40].

Family OFT-00816.

Male of Venezuelan origin diagnosed with bilateral topographic KC at age 11. The patient suffered from atopy, a history of obesity and an eye-rubbing habit whilst lacking a positive family history of KC. Before any intervention, his right eye exhibited stage 2 KC, his left eye had stage 1–2 KC, both eyes had a duck pattern, and the TA was 6,3 D in her right eye and 4,5 D in his left eye.

Several mutations in heterozygosis in genes involved in the structure of the corneal layers were detected in the patient: a missense variant (p.(Arg3362Cys)) in the *HSPG2* gene coding for perlecan; a variant affecting the splicing region (c.3635-3 C>T) in the *COL4A2* gene coding for the α2 chain of collagen IV; a missense variant (p.(Asp176Tyr)) in the *COL23A1* gene; and a variant (p.(Ile129Thr)) in the *ARG1* gene involved in the correct deposition of collagen (20). Although not all of the above genes have been directly associated with the development of KC, given the importance of arginine metabolism in the collagen secretion by corneal fibroblasts necessary for the organization of the ECM, the genes are proposed as possible candidates [52].

Finally, the presence of a missense variant in heterozygosis in the *DUXA* gene associated with corneal dystrophies such as PPCD p.(Pro40Ser) stands out for its role in development [32]. There are several case reports where variants in genes that produce PPCD have also been detected in patients with keratoconus [53]. Therefore, the *DUXA* gene is proposed as a candidate in the pathology study. The presence of several mutations in genes associated with the structure and development of the cornea accentuates the fact that KC is proposed as an oligogenic disease where, in sporadic cases, the genetic background, formed by variants inherited from both parents, contributes to its development.

Family OFT-00817.

Female of Ecuadorian origin diagnosed with forme frustre KC in her right eye at age 8. The patient presented atopy (atopic dermatitis), congenital astigmatism and amblyopia in her left eye and an overweight and obesity history, whereas she lacked a positive family history of KC and an eye-rubbing habit. Before any intervention, she had a TA of 1.4 D in her right eye and 3.2 D in her left eye.

The patient has a missense variant p.(Asp783Tyr) in heterozygosis in the *EML6* gene, which is associated with astigmatism and keratoconus [32]. Although the function of *EML6* in the eye is still unknown, the association of mutations in this gene with the above ocular phenotypes indicates that it is involved in the distribution of nutrients and the secretion of proteins that allow the maintenance of a homogeneous, rigid and avascular cornea [49]. A second variant (p.(Gly50Ala)) in heterozygosis of maternal origin is located in the *SLC4A4* gene (OMIM: #603345), which is associated with the phenotype of renal tubular acidosis with autosomal recessive ocular involvement (MIM: #604278). The gene encodes for an electrogenic sodium bicarbonate cotransporter in the basolateral membrane of corneal endothelial cells, contributing to their normal function. Mutations in the gene have been associated with increased bicarbonate concentration and, consequently, calcium accumulation in the corneal stroma, band keratopathy and corneal degeneration [33]. It may be involved in the etiology of keratoconus.

The proband has a p.(Ile374Val) missense variant in heterozygosis, inherited from the father, in the *CPSF3* gene. *CPSF3* is a critical enzyme for RNA polyadenylation, necessary to convert heteronuclear RNA to mRNA. The *CPSF3* protein is expressed in corneal epithelial and stromal cells, where it interacts with the C-terminus of the CSR1/SCARA3 isoform (a major ROS/RNS scavenger) and translocates from its location in the nucleus to a cytosolic distribution. Alterations in this gene are related to keratoconus, as they are associated with changes in cell death and alterations in other cellular functions as a consequence of the non-elimination of cells with ROS and RNS accumulation [28]. 

In addition, two variants in heterozygosis in genes coding for cadherins (*CDH23* (maternally inherited) and *CDHR1* (paternally inherited)) involved in cell adhesion necessary for corneal development have been detected [32]. Cadherins are involved in the maintenance of adequate corneal structure, and therefore, variations in the expression of cadherins have been detected in patients with keratoconus. *CDHR1* (OMIM: #609502) is related to cone and rod dystrophy and autosomal recessive RP by contributing to the maintenance of the photoreceptor structure [33]. Defects in the cell adhesion function of cadherins could be associated with KC, suggesting both genes as candidates [54].

Family OFT-00818.

Male of Spanish origin diagnosed with bilateral topographic KC at age 12. This patient had a positive family history (his father had topographic KC in his left eye and KC suspect in his right one), atopy (asthma, atopic dermatitis, multiple allergies), blepharitis, an eye-rubbing habit and the following systemic conditions: OSA and a history of obesity. Before any intervention, his right eye had stage 3 KC, his left eye had stage 2 KC, both eyes presented a duck pattern, and the TA was 4.3 D in his right eye and 2.5 D in his left eye.

In the genetic study, a missense variant p.(Glu728Asp) in heterozygosis was found in the *ZEB1* gene in both the son and the father (affected). The variant is not described in GnomAD. *ZEB1* (OMIM: #189909) is associated with FECD type 6 (MIM: #613270) and PPCD type 3 (MIM: #609141) with autosomal dominant inheritance. *ZEB1* is a nuclear gene encoding for a zinc finger transcription factor that is a transcriptional repressor that inhibits Interleukin 2 (IL-2) gene expression and represses the E-cadherin promoter at the mesenchyme–epithelial transition (MET). Although its role in the cornea is still completely unknown, mutations in *ZEB1* could favor the inflammatory process in the cornea and prevent the relevant MET process during development. Thus, articles link *ZEB1* to keratoconus. Heterozygous mutations have been found in patients with FECD or PPCD and KC and in probands with KC [53,55].

A missense variant p.(Ser306Pro) in heterozygosis in *ITGB4* classified as VUS was also found in both patients. *ITGB4* (OMIM: #147557) is associated with epidermolysis bullosa with autosomal dominant inheritance. However, this gene encodes for the ItgB4 integrin, which is the main structural protein of the hemidesmosomes that connect corneal basal epithelial cells to the basement membrane, thus being key in maintaining the integrity and function of the cornea. When corneal injury occurs, hemidesmosomes are degraded, allowing epithelial cell migration, and then rebuilt, a process regulated by the microRNA *miR-184*. Thus, mutations in *ITGB4* could be related to keratoconus. *miR-184* acts by regulating not only the expression of *ITGB4* but also that of *INPPL1* (OMIM: #600829). *INPPL1* seems to regulate, in the case of corneal injury, the rapid apoptosis of the stromal keratocytes underlying the site of injury. Thus, variants in *INPPL1* may be associated with the development of keratoconus [56]. The relevance of discussing this last gene is due to the fact that the patient presents a variant (c.2659+4G>C) in heterozygosis, de novo or maternally inherited, which affects the splicing region (verified with Alamut) and which could be the trigger for the keratoconus disease to manifest earlier and with a worse prognosis in the son than in the father.

Also, the patient has a missense variant p.(Arg452Ser) in heterozygosis in the *IDUA* gene. *IDUA* (OMIM: #25280) is related to autosomal recessive Mucopolysaccharidosis (MPS), where the enzymatic deficiency of α-L-iduronidase leads to the accumulation of glycosaminoglycans in the cells, generating a multisystemic disease that especially affects the corneal stroma. The accumulation of glycosaminoglycans in keratocytes and in the ECM leads to abnormal collagen fiber entanglement in the ECM, producing corneal opacity. Treatment with adenoviruses associated with the wild-type sequence of the *IDUA* gene (AAV-*IDUA*) in a canine model with MPS I has been shown to produce a significant therapeutic effect by inducing a phenotypic correction in the corneal stroma. Given the importance of *IDUA* in the cornea and the appearance of variants in several families in the study, it is proposed as a candidate gene in the development of KC [57].

Family OFT-00819.

Female of Honduran origin diagnosed with bilateral topographic KC at age 11. This patient had a positive family history of KC (her mother had bilateral KC suspect) and a history of being overweight. She did not report atopy or an eye-rubbing habit. Before any intervention, both eyes had stage 1 KC; however, her right eye had a croissant pattern and her left one had a bowtie pattern, and the TA was 1.8 D and 2.6 D, respectively.

A missense variant p.(Gly1195Val) in heterozygosis in the *INPPL1* gene was found in both affected patients (mother and daughter). Given the involvement of *INPPL1* in the apoptosis of stromal keratocytes exposed in the proband OFT-00818, it is suggested as a possible candidate gene in the etiology of keratoconus [56]. Likewise, a missense variant p.(Arg574Cys) in heterozygosis in the *GUCY2D* gene was found in the mother and daughter. *GUCY2D* (OMIM: #600179) is related to central areolar choroidal dystrophy (MIM: #215500) with autosomal dominant inheritance and cone and rod dystrophy type 6 (MIM: #601777) and Leber congenital amaurosis (LCA) (MIM: #204000) with autosomal recessive inheritance, among other pathologies. *GUCY2D* encodes for a retina-specific guanylate cyclase involved in visual phototransduction and has been associated with causing KC in patients with LCA [32]. Furthermore, two variants in the *CDH23* and *CDHR1* genes that appear altered in the OFT-00817 family have been identified in both affected patients.

Family OFT-00820.

Male of Colombian origin diagnosed with bilateral topographic KC at age 13. This patient presented a positive family history of KC (his mother had bilateral KC suspect), an eye-rubbing habit, scoliosis and the following ocular conditions: posterior polymorphous corneal dystrophy (PPCD), retinopathy of prematurity and blepharitis. He did not suffer from atopy. Before any intervention, his right eye had stage 3 KC and a duck pattern, and his left eye had a KC-suspected cornea; the corneal topographic astigmatism (TA) was 7.8 D in his right eye and 1 D in his left eye.

A missense variant p.(Arg3159Gln) in heterozygosis was detected in both affected individuals (father and son) in the *HSPG2* gene, whose relevance in KC is described in families OFT-00813 and OFT-00816. Another missense variant in the *IMPG2* gene (OMIM: #607056) associated with autosomal dominant macular dystrophy type 5 (MIM: #616152) and autosomal recessive Retinitis Pigmentosa (OMIM: #613581) was found in both affected individuals. The gene encodes for a proteoglycan that binds to chondroitin sulfate and hyaluronic acid, contributing to the organization of the interphotoreceptor matrix [32]. Although it is generally associated with retinal diseases, the association of keratoconus with these diseases and the role of the gene lead to its proposal as a candidate [4]. 

The proband also presents a variant in heterozygosis with paternal inheritance affecting the splicing region in the *CPMD8* gene (OMIM: #608841), which is involved in eye development and is associated with developmental anomalies of anterior segment 8 (MIM: #617319), such as corneal defects and the abnormal migration of neural crest cells derived from mesenchymal cells, affecting the development of the posterior surface of the cornea [32,33]. Given the relevance of the gene in corneal development, the variant in the proband could enhance KC progression.

Family OFT-00821.

Male of Spanish origin diagnosed with bilateral topographic KC at age 15. This patient had atopy (asthma, multiple allergies) and an eye-rubbing habit and did not report a positive family history of KC. Before any intervention, his left eye showed stage 2 KC, whilst his left eye had stage 1–2 KC; a snowman pattern was present in both of his eyes, and the TA was 3 D in his right eye and 1.2 D in his left eye.

The patient’s genotype features a missense variant p.(Glu751Gln) in heterozygosis of maternal inheritance in the *INPPL1* gene. Its possible involvement in keratoconus has already been described in the OFT-00818 and OFT-00819 families [56]. The patient also has a variant missense p.(Ser269Cys) in heterozygosis of paternal inheritance in the *IDUA* gene, whose association with the etiology is described in the OFT-00818 family [57].

Family OFT-00846.

Female of Dominican origin diagnosed with bilateral topographic KC at age 10. This patient had a positive family history of KC (her mother had bilateral form fruste KC), atopy (hay fever, allergic conjunctivitis) and an eye-rubbing habit. Before any intervention, her left eye had stage 1 KC and a snowman pattern, and her left eye had stage 2–3 KC and a duck pattern; the TA was 5.1 D and 10 D, respectively.

Among the genetic variants found, a missense variant p.(Gly1089Arg) in heterozygosis with maternal inheritance in the *ITGB4* gene, which is mutated in the OFT-00818 family, stands out, where its possible implication in the pathology is exposed [56]. Similarly, a missense variant in heterozygosis, also maternally inherited, in the *SPARC* gene is detected in the patient’s genotype. *SPARC* (OMIM: #182120) encodes for a protein that is expressed during embryogenesis and in the remodeling or repair of adult tissues by participating in processes of cell–cell and cell–matrix interactions, differentiation, the production and organization of the ECM, wound healing and angiogenesis. Although OMIM links the gene to autosomal recessive osteogenesis imperfecta, missense variants in *SPARC* have been detected in patients with keratoconus, so the involvement of the gene in the pathology is possible. In addition, the proximal location of *SPARC* to the *LOX* gene, residing on chromosomes 5q31.3-q32 and 5q23.2, respectively, could also suggest its contribution to keratoconus [58]. A third variant in heterozygosis appears in the *TTN* gene (p.(Val24529Leu)), whose relationship with KC is argued in the OFT-00814 family [40].

Family OFT-00847.

Female of Spanish origin diagnosed with bilateral KC suspect at age 8. This patient reported a positive family history of KC (her father had KC suspect in his left eye, and her uncle was diagnosed with KC), while she lacked atopy, an eye-rubbing habit or any other eye or systemic history of interest. Before any intervention, she had a TA of 4.2 D in her right eye and 2.2 D in her left eye.

Both affected individuals (father and son) present two missense variants in heterozygosis, p.(Arg25Ser) and p.(Gly305Ser) in *KRT7* (OMIM: #148059) and *KRT19* (OMIM: #148020) genes. These genes, respectively, encode for cytokeratin 7 and cytokeratin 19, which are essential during the differentiation of the corneal epithelium and endothelium. OMIM reports that changes in the expression of both cytokeratins have been observed in the abnormal corneal endothelium of patients with PPCD [33]. Given their relevance in corneal development and the relationship of the genetic background of PPCD with keratoconus, both genes are proposed as possible genes involved in the progression of the disease.

Family OFT-00848.

Male of Spanish and Portuguese origin diagnosed with bilateral topographic KC at age 11. This patient reported a positive family history of KC (his father was diagnosed with bilateral topographic KC), atopy (asthma, hay fever) and a history of overweight and obesity. Before any intervention, his right eye had stage 3 KC, his left eye had stage 1 KC, both of his eyes presented a duck pattern, and the TA was 5.8 D in his right eye and 1 D in his left eye.

A missense variant p.(Arg221His) in heterozygosis in the *APEX1* gene was detected in both affected individuals (father and son). *APEX1* (OMIM: #107748) codes for an APEX1 nuclease that repairs DNA following oxidative damage in the cornea through base excision repair. Two variants in *APEX1* are associated with a risk of keratoconus [59]. Given the function and existence of variants in *APEX1* associated with the keratoconus phenotype, this gene could be a good candidate causal gene. In addition, a variant insertion in a conserved region of the *IDUA* gene p.(Leu11_Ala12insAlaLeuLeu) in heterozygosis was found in both patients (father and son). The possible involvement of *IDUA* in keratoconus is presented in families OFT-00818 and OFT-00821.

Family OFT-00850.

Male of Colombian and Spanish origin diagnosed with bilateral topographic KC at age 13. This patient had a positive family history (his mother had bilateral topographic KC), atopy (asthma, atopic dermatitis, multiple allergies and allergic conjunctivitis), blepharitis and a history of obesity. However, he did not mention an eye-rubbing habit. Before any intervention, his right eye had stage 3–4 KC and a snowman pattern, and his left eye had stage 3 KC and a croissant pattern; the TA was 5 D and 2.7 D, respectively.

In the mother and son (both with KC), a missense variant p.(Gly1798Arg) in heterozygosis was detected in the *COL6A5* gene (OMIM: #611916), which codes for the α-5 chain of collagen VI, which constitutes one of the collagens of the ECM of the corneal stroma. Variants in this gene have been described in KC probands [1,60]. Missense variants in heterozygosis were found in both *GRHL1* (p.(Val287Gly)) and *SLC4A4* (p.(Gly50Ala)), causally related to corneal dystrophies. *GRHL1* (OMIM: #609786) codes for an essential transcription factor in the ETM transition of corneal development. Mutations in this gene are mainly associated with PPCD; however, given the function of the factor for which it codes, its contribution to KC could be suggested [32,33]. The function and possible involvement of *SLC4A4* in KC is presented in the OFT-00817 family.

Family OFT-00851.

Female of Spanish origin diagnosed with bilateral topographic KC at age 16. This patient presented atopy and an eye-rubbing habit whilst lacking a positive family history of KC and any other medical conditions of interest. Before any intervention, her right eye had stage 2 KC and a duck pattern, and her left eye had stage 3–4 KC and a nipple pattern; the TA was 4.4 D and 0.9 D, respectively.

In the genetic study of the proband, a missense variant in heterozygosis was found in the *EML6* gene, directly related to keratoconus, as described in family OFT-00817. The patient also has a missense variant p.(Arg246Trp) in heterozygosis in the *COL9A3* gene. *COL9A3* (OMIM: #120270) encodes for the α3 chain of collagen XI and is associated with Stickler syndrome (MIM: #620022) with autosomal recessive inheritance. Heterozygous mutations in this gene have been found in patients with keratoconus by contributing to stromal ECM disruption [1,61]. In addition, the patient presents, like the probands of the OFT-00814, OFT-0081 and OFT-00846 families, a variant in *TTN*, whose relationship with KC has been previously documented [40].

Family OFT-00852.

Male of Honduran origin diagnosed with bilateral topographic KC at age 18. This patient did not report a positive family history of KC, an eye-rubbing habit, atopy or any other medical conditions of interest. Before any intervention, his right eye had stage 2 KC and a duck pattern, whereas his left eye had stage 3–4 KC and a nipple pattern; the TA was 3.8 D and 4.7 D, respectively.

The genotype of this proband includes a missense variant p.(Gly769Arg) in heterozygosis in the *SLC4A11* gene, described in GnomAD to be associated with corneal dystrophy. *SLC4A11* (OMIM: #610206) is directly related to keratoconus, as it codes for a bicarbonate and sodium transporter that regulates oxidative stress in the corneal endothelium by enhancing antioxidant defenses [1,32,33]. Also, the proband presents a missense variant (p.(Tyr4328Cys)) in heterozygosis in *LRP1B*. *LRP1B* (OMIM: #608766) is responsible for cholesterol regulation through clathrin-mediated endocytosis, lysosomal transport and degradation. Thus, in the cornea, it is associated with lysosomal degradation and the removal of lipid by-products associated with oxidative stress and with the response to corneal injury and inflammation. Four variants in *LRP1B* appear to be the cause of keratoconus development in four families. A missense variant (p.(Glu284Lys)) in heterozygosis also appears in another gene involved in corneal inflammatory and injury response processes, *PIK3CG*. *PIK3CG* (OMIM: #601232) encodes for phosphoinositol-3-kinase, which acts in the Wnt and Akt pathway, regulating the collagen lattice and corneal immunity. Three variants in *PIK3CG* have been described in families affected by keratoconus [49].

Furthermore, another missense variant, p.(Tyr1833His), in heterozygosis was found in *COL6A6* (OMIM: #616613), which encodes the α6 chain of collagen VI. Although the most prevalent collagens in the corneal stroma are types I and V, collagen VI appears to be relevant in the maintenance of ECM organization, and, therefore, this gene could be involved in the pathophysiology [45]. The patient also presents a missense variant in heterozygosis in *LOXHD1*, as reported in the OFT-00242 and OFT-00814 families.

Family OFT-00853.

Male of Spanish origin diagnosed with bilateral topographic KC at age 14. This patient had a positive family history of KC (his mother had KC suspect in her right eye, his father had bilateral KC suspect, and his maternal aunt had KC suspect in both eyes), as well as atopy (atopic dermatitis) and a history of being overweight. He also presented an eye-rubbing habit. Before any intervention, his right eye had stage 2 KC, his left eye had stage 2–3 KC, a duck pattern was present in both of his eyes, and he had a TA of 4.8 D in his right eye and 4–6 D in his left eye.

The patient has two paternally inherited missense variants in heterozygosis in the *LRP1B* (p.(His856Gln)) and *CDHR1* (p.(Asn466Ser)) genes. Both genes have been proposed as genes involved in KC in the OFT-00852 and OFT-00818 and OFT-00819 families, respectively.

Another missense variant p.(Tyr58Cys) in heterozygosis with maternal inheritance was detected in the *ALDH1A2* gene. *ALDH1A2* (OMIM: #603687) translates into an aldehyde dehydrogenase involved in the production of retinoic acid (RA). A deficiency in RA has been described to affect stromal integrity and lead to the development and progression of corneal diseases such as keratoconus. Animal models with Aldh1a2 protein deficiency are associated with corneal defects due to a reduced corneal stroma and corneal epithelium. The presence of the mutated gene in both patients and the results of functional models may suggest *ALDH1A2* as a possible candidate gene [62].

Family OFT-00854.

Male of Spanish origin diagnosed with topographic KC in his left eye at age 17. This patient had a positive family history (his sister had KC suspect bilaterally), atopy (atopic dermatitis) and a congenital cataract in his left eye. However, he did not report an eye-rubbing habit or any systemic condition of interest. Before any intervention, his left eye had stage 3 KC, a duck pattern and a TA of 5.3 D, whilst his right eye’s TA was 0.8 D. 

A missense variant p.(Ala310Val) in heterozygosis was detected in the *IPO5* gene (OMIM: #602008) with maternal inheritance. *IPO5* encodes for a β-importin. The β-importins interact with nucleoporins, allowing the translocation of the complex (alpha importins bound to nuclear localization signal (NLS) proteins). Although their involvement in the development of keratoconus is unknown, gene variants have been described in patients from the Chinese and Polish populations [1,45]. 

The proband also has another maternally inherited missense variant in heterozygosis in the *GALNT14* gene. *GALNT14* transfers the N-acetyl-D-Galactosamine residue to serine or threonine residues on substrate proteins, especially mucins, catalyzing the initial step for O-oligosaccharide biosynthesis. Ocular diseases have been linked to failures in mucin glycosylation. In particular, one article points to a frameshift variant in homozygosity in the *GALNT14* gene as a cause of non-syndromic keratoconus. The SIFT-score value of the variant is 0.002, and, in addition, Uniprot predicts that the variant produces a deleterious protein, as the catalytic subdomain A is affected. Although the variant in the proband is found in heterozygosis, the mutation could contribute to an abnormal protein function and to the progression of KC [63].

A third missense variant p.(Gly549Ser) in heterozygosis appears in the *COL8A2* gene, already proposed as a candidate gene in the OFT-00273 family for its contribution to the ECM.

Family OFT-00855.

Male of Spanish origin diagnosed with bilateral topographic KC at age 13. This patient had an eye-rubbing habit, blepharitis and a history of obesity. However, he did not report a positive family history or atopy. Before any intervention, his right eye had stage 3 KC, his left eye had stage 3–4 KC, the TA was 10 D and 8.8 D, respectively, and both eyes presented a nipple pattern.

In the patient’s genotype, a missense variant p.(Arg1323Cys) in heterozygosis is found in *ADAMTS9* (OMIM: #605421), which codes for an enzyme that cleaves large proteoglycans such as aggrecan and versican and is involved in the transport of secretory elements from the endoplasmic reticulum to the Golgi apparatus. This gene is located in chromosomal region 3p14.3-p14.2, which has been reported to generate autosomal dominant keratoconus type 3 (MIM: #608586). Thus, variants associated with keratoconus have been detected in *ADAMTS9* [33]. Another missense variant p.(Gly236Glu) in heterozygosis was found in another gene coding for a metalloproteinase of the ADAMTS family, *ADAMTS8* (OMIM: #605175), which has been described to be associated with central corneal thinning and keratoconus. Although the involvement of the gene in the etiology is unknown, it appears to be involved in the biomechanical properties of the cornea [64].

Family OFT-00856.

Female of Spanish and Mexican origin diagnosed with topographic KC in her left eye at age 14. This patient had an eye-rubbing habit, blepharitis and the following systemic conditions: scoliosis and a history of being overweight. She did not report a positive family history or atopy. Before any intervention, her left eye had stage 1–2 KC, a duck pattern and a TA of 9.8 D, whereas the TA in her right eye was 0.7 D. 

Missense variants in heterozygosis in genes described to be associated with keratoconus were detected in the proband: *EML6*, which appears in the OFT-00817 and OFT-00851 families and whose implication has been previously reported, and *DOCK9* (OMIM: #607325). *DOCK9* is involved in cadherin binding [1,45]. It is a gene that presents incomplete penetrance; thus, a variant in *DOCK9* was detected in the affected and healthy members of a family with autosomal dominant keratoconus type 7 (MIM: #614629) [33]. Likewise, another missense variant was detected in *ADAMTS7* (OMIM: #605009), which is a gene encoding for a metalloproteinase of the ADAMTS family. Given that these metalloproteinases play a relevant role in ECM proteolysis and that other proteins of the family (such as ADAMTS9) have been associated with KC, this gene could contribute to the genetic background of the pathology [64]. Collagen V is the most prevalent collagen in the cornea, so the presence of a missense variant in heterozygosis in the *COL5A3* gene (OMIM: #120216), which codes for the α3 chain of collagen V, stands out in the patient’s exome [12]. Given the functional importance of this gene, it is proposed as a gene involved in the development of keratoconus.

Family OFT-00857.

Male of Spanish origin diagnosed with bilateral topographic KC at age 14. This patient suffered from atopy (asthma, rhinoconjunctivitis, atopic dermatitis, allergic conjunctivitis, multiple allergies), an eye-rubbing habit and a history of obesity; he also reported a positive family history of KC (his maternal uncle and two of his mother’s cousins had KC). Before any intervention, his right eye had stage 2 KC, his left eye had stage 1 KC, the TA was 4.8 D and 2 D, respectively, and both of his eyes presented a snowman pattern.

A missense variable p.(Arg492Gln) in heterozygosis was detected in *ZNF469*, described in GnomAD for autosomal recessive Ehlers–Danlos syndrome and unassociated with the phenotype. The presence of mutations in this gene also in the OFT-00242 and OFT-00813 families, the association of keratoconus with the Ehlers–Danlos genetic background and the literature linking this gene to the development of KC make it an important candidate in the pathophysiology studied [38]. 

The proband presents a missense variant (p.(Arg452Ser)) in heterozygosis in the *IDUA* gene with maternal inheritance. With the action of *IDUA* in the cornea described in families OFT-00818, OFT-80021 and OFT-00848, the gene is proposed as a possible candidate gene. 

Another missense variant with maternal inheritance in heterozygosis in the *LOXL1* gene also appears in the proband. *LOXL1* (OMIM: #153456) encodes for a lysyl oxidase involved in elastin production and ECM maintenance. Given the relevance of the functional deficiency of LOX enzymes in keratoconus, *LOXL1* is proposed as a possible candidate gene [1,45].

To conclude, the study of the exomes of these patients shows that non-syndromic keratoconus is rarely caused by a single mutation, but it seems to be a disease with oligogenic inheritance. Thus, the pathology is the result of a set of altered genes that, through the functions they perform, lead to the phenotype studied. In sporadic cases, it appears that the genetic background formed by variants inherited from both parents results in the predisposition to develop the disease. On the other hand, although the heritability of keratoconus evidences the importance of the genetic background in the disease, it is necessary to point out that the environments and habits of the subjects seem to be relevant in the clinical characteristics, progression and prognosis of keratoconus. 

Variants were detected in genes described for other associated pathologies (*PIKFYVE*, *ZNF469*, *LOXHD1*, *GUCY2D*, *SLC8A3*, *DUXA*) and in genes involved in the development (*ZEB1*, *CPMD8*, *KRT7*, *KRT9*, *GRHL1*), structure (*MMP1*, *MMP2*, *LOXL1*, *LOXL2*, *TNXB*, *HSPG2*, *ADAMTS9*, *ADAMTS7*, *AD*-*AMTS8*, *IMPG2*, *SPARC*, *COL2A1*, *COL4A2*, *COL4A3*, *COL4A4*, *COL5A3*, *COL6A3*, *COL6A5*, *COL6A6*, *COL8A2*, *COL9A3*, *COL23A1*), cell adhesion (*CDH23*, *CDHR1*, *DOCK9*), functions (*NDRG1*, *TTN*, *EML6*, *SLC4A4*, *PPIP5K2*, *PIK3CG*, *LRP1B*, *GALNT14*, *ALDH1A2*, *IPO5*), oxidative stress regulation (*CPSF3*, *APEX1*, *SLC4A11*) and repair (*ITGB4*, *INPPL1*, *IDUA*) of the cornea.

Of the genes suggested in this article as likely causative of keratoconus, 63% had already been linked in at least one article to the disease, and 37% are proposed as possible genes involved in its development (*GUCY2D*, *SLC8A3*, *DUXA*, *CPMD8*, *KRT7*, *KRT9*, *GRHL1*, *CDH23*, *CDHR1*, *SLC4A4*, *ALDH1A2*, *CPSF3*, *ADAMST7*, *IMPG2*, *COL23A1*, *ITGB4*, *INPPL1*, *IDUA*). 

Most patients have a group of genetic variants that jointly result in the development of keratoconus. Clusters of genes that affect the organization of the corneal structure are detected, as in the case of the OFT-00815 family. Likewise, variants in genes involved in different processes implicated in the physiopathology of keratoconus were detected in families, as is the case of the OFT-00818 family. In the OFT-00818 family, variants in genes affecting corneal development, corneal structural organization and corneal repair were detected.

Although most cases are of oligogenic inheritance, the OFT-00290 family could suggest that, in isolated cases, a single gene may contribute in a special way to the development of the disease. This would be interesting from a therapeutic perspective, as there are already gene therapies targeting genes involved in eye diseases, such as *RPE65* or *RPGR*, associated with an increased field of vision in patients [65].

The study itself has the limitations of small sample size and a lack of functional validation of the proposed variants, as well as the limitation associated with the difficulty of variant selection due to the low number of articles on the genetics of the disease and the oligogenic nature of the disease. Nevertheless, the findings of this NGS study may be of use in elucidating the pathophysiology of the disease and serve as a reference for future genetic studies in patients with keratoconus.

To corroborate the involvement of these genes in pediatric keratoconus, further research is needed through segregation studies (if they are necessary) and functional studies to verify the functional effects of the variants detected.

## 5. Conclusions

Knowledge of the genes involved in keratoconus is essential to understanding the disease and to proposing new therapeutic targets for this corneal dystrophy, which currently does not have an effective treatment in all cases. For this study of patients with keratoconus, the coordinated work of multidisciplinary teams has been essential to improving patient care and diagnosis. 

The study of the NGS results of the exomes of pediatric patients suggests that keratoconus is a disease of oligogenic inheritance, where the cumulative effect of different variants of uncertain significance in genes involved in the development, structure, cell adhesion, functions and repair pathways in the cornea lead to the pathology. 

Further studies are needed to confirm the involvement of the candidate genes described in this article in the development of pediatric keratoconus. As possible future lines of work, we suggest the study of segregation in patients who have not been studied with their progenitors to determine whether the variants that they present are de novo. In addition, in patients with a positive first-degree family history of the pathological phenotype of keratoconus, the genetic study could be extended to a genomic study. The study of protein expression in the cornea (if any patient undergoes corneal transplantation) and the study of transcripts through the use of minigenes are also proposed to identify the functional implications of the genetic variants.

The findings of this NGS study may be useful in elucidating the altered physiological processes involved in the development of keratoconus and may be useful for future genetic studies in patients with the disease.

## Figures and Tables

**Figure 1 genes-14-01838-f001:**
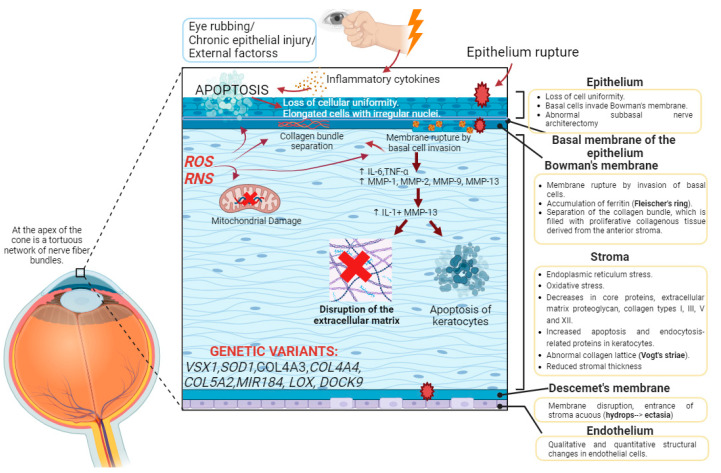
Cascade of molecular processes in the pathophysiology of keratoconus. Summary of the molecular involvement of the corneal layers. The presence of external factors, such as eye rubbing, increased reactive oxygen species (ROS), mitochondrial damage or mutations in certain genes, can lead to the activation of biochemical cascades that first induce the loss of cellular uniformity in the corneal epithelium. The process continues with the inflow of epithelial cells into Bowman’s membrane, leading to its rupture and the accumulation of ferritin in the area, which clinically translates into the clinical sign of Fleischer’s ring. There is an increase in proinflammatory cytokines (IL-6, TNF-alpha) and an exacerbation of the function of metalloproteinases (MMP-1, MMP-2, MMP-9, MMP-13), resulting in the separation of collagen bundles and their abnormal arrangement (clinically perceived as Vogt’s striae), increased apoptosis of keratocytes, leading to a reduction in corneal stromal thickness, and both histological changes and disruption of the extracellular matrix. Created with Biorender.com (accessed on 29 August 2023).

**Figure 2 genes-14-01838-f002:**
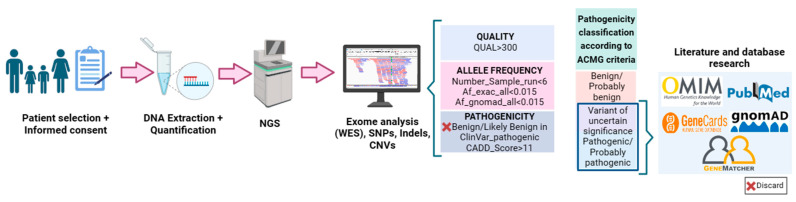
Workflow for the methodology of exome analysis in families of keratoconus patients. Created with Biorender.com (accessed on 29 August 2023).

**Figure 3 genes-14-01838-f003:**
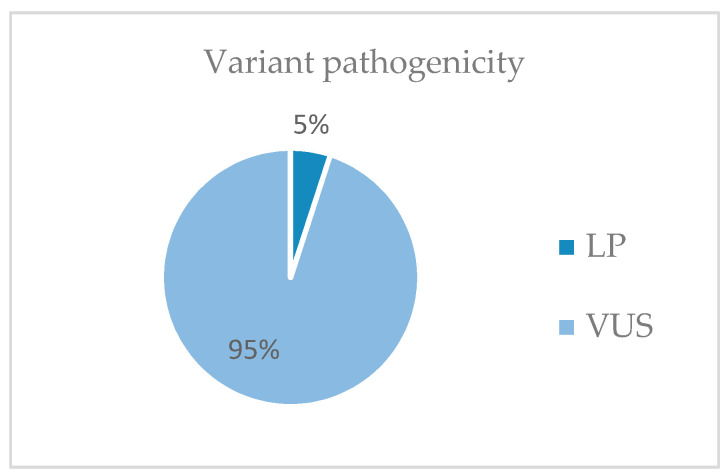
Pathogenicity of the variants detected in the study. LP, likely pathogenic; VUS, variant of uncertain significance.

**Figure 4 genes-14-01838-f004:**
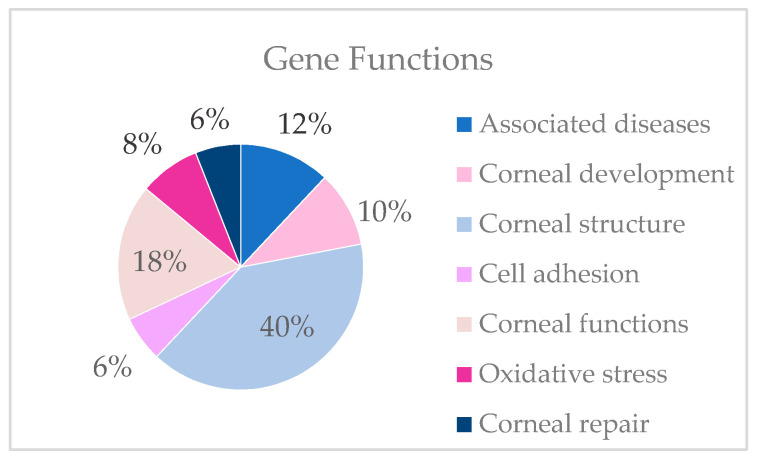
Functions of the genes selected as candidates for KC in which variants were detected in the study.

**Table 1 genes-14-01838-t001:** Criteria for classification of keratoconus stages by corneal topography using OCULUS PENTACAM^®^ Standard Software Program.

Grade	Visual Acuity with Glasses	Visual Acuity with CL ^1^	Corneal Indices		Ecc. 30 ^4^	RMin ^5^	Cornea
			ISV ^2^	KJ ^3^			
Suspect	1.0 1.0–1.25 20/15	1.0 1.0–1.25 20/15	<30	1.04 1.04–1.07 1.07	4 normal values	7.8 7.8–6.7 6.7	Clear, inconspicuous cornea. Horizontal, oval or with circular shadows centered or slightly off-center.
Grade 1	20/25 0.8–1.25 20/15	1.0	30 30/55 55	1.07 1.07–1.15 1.15	At least 1 value is rarely abnormal	7.5 7.5–6.5 6.5	Transparent cornea. Fleischer’s ring at the base of the apex. Decrease in the apex vertex.
Grade 2	20/60 0.32–1.0 1.0	20/30 0.63–1.0 1.0	55 55–90 90	1.10 1.10–1.25 1.25	At least 1 value is frequently abnormal	6.9 6.9–5.3 5.3	Cornea often remains transparent. The apex occasionally becomes somewhat decentered. Partial or circular Fleischer rings. Occasionally, Vogt’s striae are present.
Grade 3	20/125 0.16–0.63 20/30	20/40 0.5–1.0 1.0	90 90–150 150	1.15 1.15–1.45 1.45	1 value is always abnormal	6.6 6.6–4.8 4.8	The apex has decreased in thickness, is off-center and is slightly dark. Fleischer rings clearly visible and circular. Easily identifiable Vogt’s striae. Eventually signs of Munson.
Grade 4	<20/400 <0.05–0.2 20/100	20/100 0.2–0.5 20/40	>150	>1.50	1 value is always abnormal	<5	The cornea is often scarred. Munson’s sign is evident.

^1^ Contact lenses (CLs); ^2^ index of surface variation (ISV); ^3^ keratoconus index (KI); ^4^ eccentricity value at an angle of 30º of the front cornea (Ecc. 30); ^5^ minimum value of corneal surface curvature (RMin).

**Table 2 genes-14-01838-t002:** Clinical characteristics of the probands: classification of KC according to grade and morphological phenotype; sex and age at proband diagnosis; family history of KC; information on who is undergoing the genetic study.

Family ID	Keratoconus				Sex	Age	Family History	Genetic Study
	RE ^1^		LE ^2^					
	Grade	Morphology	Grade	Morphology				
OFT-00242	1	Snowman	1–2	Croissant	M ^3^	15	No	Proband
OFT-00273	2–3	Snowman	2	Snowman	M	13	No	Proband
OFT-00290	1–2	Duck	3–4	Snowman	M	14	Mother with FCD ^5^.	Proband + mother
OFT-00705	1–2	Duck	2	Duck	M	13	Father with KC ^6^ BE ^7^.	Proband + parents
OFT-00813	1	Duck	Frustre	X	M	10	Father with suspect KC RE.	Proband + parents
OFT-00814	3	Duck	1	Duck	M	17	No	Proband + parents
OFT-00815	3	Nipple	1	Bowtie	M	14	Mother with KC BE. Grandmother and maternal uncles with astigmatism.	Proband + parents
OFT-00816	2	Duck	1–2	Duck	M	11	No	Proband + parents
OFT-00817	Frustre	-	-	-	F ^4^	8	No	Proband + parents
OFT-00818	3	Duck	2	Duck	M	12	Father with suspect KC RE and topographic LE.	Proband + father
OFT-00819	1	Croissant	1	Bowtie	F	11	Mother with suspect KC BE.	Proband + mother
OFT-00820	3	Duck	Suspect	Snowman	M	13	Mother with suspect KC BE.	Proband + parents
OFT-00821	2	Snowman	1–2	Snowman	M	15	No	Proband + parents
OFT-00846	1	Snowman	2–3	Duck	F	10	Mother with frustre KC BE.	Proband + parents
OFT-00847	Suspect	Croissant	Suspect	Croissant	F	8	Father with suspect KC LE. Paternal uncle with topographic KC.	Proband + parents
OFT-00848	3	Duck	1	Duck	M	11	Father with frustre KC BE.	Proband + parents
OFT-00850	3–4	Snowman	3	Duck- Croissant	M	13	Mother with suspect KC BE.	Proband + parents
OFT-00851	3	Croissant	1	Duck	F	16	No	Proband + parents
OFT-00852	2	Duck	3–4	Nipple	M	18	No	Proband
OFT-00853	2	Duck	2–3	Duck	M	14	Father with suspect KC BE, mother with suspect KC LE. Maternal cousin with topographic KC BE.	Proband + parents
OFT-00854	-	-	3	Duck	M	17	Sister with suspect KC.	Proband + parents
OFT-00855	3	Nipple	3–4	Nipple	M	13	No	Proband
OFT-00856	-	-	1–2	Duck	F	14	No	Proband + parents
OFT-00857	2	Snowman	1	Snowman	M	14	Maternal uncle with KC.	Proband + mother

^1^ RE: right eye; ^2^ LE: left eye; ^3^ M: male; ^4^ F: female; ^5^ FCD: Fleck Corneal Dystrophy; ^6^ KC: keratoconus; ^7^ BE: both eyes.

**Table 3 genes-14-01838-t003:** Gene variants found in each family (“OFT”) in the study of keratoconus in pediatric patients.

Family	Gene	Mutation	ACMG * Result	Zygosity	Inheritance	De Novo/Inherited
OFT-00242	*ZNF469*	NM_001127464.2:c.11173G>A p.(Ala3725Thr)	VUS ^1^	Het ^3^	AR ^4^	Unknown
	*NDRG1*	NM_001135242.1:c.699-2A>G	LP ^2^	Het	AR	Unknown
	*LOXHD1*	NM_144612.6:c.1843C>T p.(Arg615Trp)	VUS	Het	AR	Unknown
OFT-00273	*MMP1*	NM_002421.3:c.1301G>T p.(Gly434Val)	VUS	Het	Unknown	Unknown
	*MMP2*	NM_004530.5:c.586G>A p.(Ala196Thr)	VUS	Het	AR	Unknown
	*COL8A2*	NM_005202.3:c.567A>T p.(Glu189Asp)	VUS	Het	AR	Unknown
	*COL4A3*	NM_000091.4:c.4523A>G p.(Asn1508Ser)	VUS	Het	AR	Unknown
OFT-00290	*PIKFYVE*	NM_015040.3:c.3791+2T>C	LP	Het	AD ^5^	Maternal
	*LOXL2*	NM_002318.2:c.1642C>A p.(Pro548Thr)	VUS	Het	Unknown	Maternal
OFT-00705	*PPIP5K2*	NM_015216.4:c.86A>C p.(Asp29Ala)	VUS	Het	Unknown	Paternal
	*SLC8A3*	NM_182932.2:c.56T>C p.(Val19Ala)	VUS	Het	Unknown	Paternal
OFT-00813	*ZNF469*	NM_001127464.2:c.9347C>T p.(Thr3116Ile)	VUS	Het	AR	Paternal
	*TNXB*	NM_019105.6:c.4010G>A p.(Arg1337His)	VUS	Het	AD/AR	Paternal
	*HSPG2*	NM_005529.6:c.2023C>T p.(Arg675Trp)	VUS	Het	AD/AR	Maternal
OFT-00814	*LOXHD1*	NM_144612.6:c.740C>A p.(Ala247Glu)	VUS	Het	AR	Paternal
	*TTN*	NM_133378.4:c.91524G>C p.(Glu30508Asp)	VUS	Het	AD	Maternal
OFT-00815	*COL6A3*	NM_004369.3:c.8865dupC p.(Ala2956fs)	LP	Het	AD/AR	Paternal
	*COL6A3*	NM_004369.3:c.5681C>T p.(Pro1894Leu)	VUS	Het	AD/AR	Maternal
	*COL4A4*	NM_000092.4:c.4291C>T p.(Arg1431Cys)	VUS	Het	AD/AR	Maternal
	*COL2A1*	NM_033150.2:c.402G>A p.(Met134Ile)	VUS	Het	AD	Maternal
	*TTN*	NM_133378.4:c.64442T>C p.(Leu21481Pro)	VUS	Het	AD	Maternal
OFT-00816	*HSPG2*	NM_005529.6:c.10084C>T p.(Arg3362Cys)	VUS	Het	AD/AR	Maternal
	*COL4A2*	NM_001846.3:c.3635-3C>T	VUS	Het	AD	Maternal
	*COL23A1*	NM_173465.3:c.526G>T p.(Asp176Tyr)	VUS	Het	Unknown	No Maternal
	*ARG1*	NM_000045.3:c.386T>C p.(Ile129Thr)	VUS	Het	AR	Maternal
	*DUXA*	NM_001012729.1:c.118C>T p.(Pro40Ser)	VUS	Het	Unknown	No Maternal
OFT-00817	*EML6*	NM_001039753.2:c.2347G>T p.(Asp783Tyr)	VUS	Het	Unknown	Maternal
	*SLC4A4*	NM_001098484.2:c.1256C>T p.(Thr419Met)	VUS	Het	AD/AR	Maternal
	*CPSF3*	NM_016207.3:c.1120A>G p.(Ile374Val)	VUS	Het	Unknown	Paternal
	*CDH23*	NM_022124.5:c.7834G>T p.(Val2612Leu)	VUS	Het	AD/AR	Maternal
	*CDHR1*	NM_033100.3:c.1133G>A p.(Arg378Gln)	VUS	Het	AR	Paternal
OFT-00818	*ZEB1*	NM_030751.5:c.2184A>C p.(Glu728Asp)	VUS	Het	AD	Paternal
	*ITGB4*	NM_001005619.1:c.916T>C p.(Ser306Pro)	VUS	Het	AD/AR	Paternal
	*INPPL1*	NM_001567.3:c.2659+4G>C	VUS	Het	AR	No Paternal
	*IDUA*	NM_000203.4:c.299+2092C>T	VUS	Het	AR	Paternal
OFT-00819	*INPPL1*	NM_001567.3:c.3584G>T p.(Gly1195Val)	VUS	Het	AR	Maternal
	*GUCY2D*	NM_000180.3:c.1720C>T p.(Arg574Cys)	VUS	Het	AD/AR	Maternal
	*CDH23*	NM_022124.5:c.7351A>G p.(Asn2451Asp)	VUS	Het	AD/AR	Maternal
	*CDHR1*	NM_033100.3:c.931C>A p.(Leu311Ile)	VUS	Het	AR	Maternal
OFT-00820	*HSPG2*	NM_005529.6:c.9476G>A p.(Arg3159Gln)	VUS	Het	AD/AR	Maternal
	*IMPG2*	NM_016247.3:c.3688G>T p.(Ala1230Ser)	VUS	Het	AR	Maternal
	*CPAMD8*	NM_015692.2:c.4914+2_4914+3insT	LP	Het	AR	Paternal
OFT-00821	*INPPL1*	NM_001567.3:c.2251G>C p.(Glu751Gln)	VUS	Het	AR	Maternal
	*IDUA*	NM_000203.4:c.806C>G p.(Ser269Cys)	VUS	Het	AR	Paternal
OFT-00846	*ITGB4*	NM_001005619.1:c.3265G>C p.(Gly1089Arg)	VUS	Het	AD/AR	Maternal
	*SPARC*	NM_003118.3:c.280G>A p.(Val94Met)	VUS	Het	AD/AR	Maternal
	*TTN*	NM_133378.4:c.73585G>C p.(Val24529Leu)	VUS	Het	AD	No Maternal
OFT-00847	*KRT7*	NM_005556.3:c.73C>A p.(Arg25Ser)	VUS	Het	Unknown	Paternal
	*KRT19*	NM_002276.4:c.913G>A p.(Gly305Ser)	VUS	Het	Unknown	Paternal
OFT-00848	*IDUA*	NM_000203.4:c.30_31insCTGGCGCTC p.(Leu11_Ala12insAlaLeuLeu)	VUS	Het	AR	Paternal
	*APEX1*	NM_001641.3:c.662G>A p.(Arg221His)	VUS	Het	Unknown	Paternal
OFT-00850	*COL6A5*	NM_001278298.1:c.5392G>A p.(Gly1798Arg)	VUS	Het	Unknown	Maternal
	*GRHL1*	NM_198182.2:c.860T>G p.(Val287Gly)	VUS	Het	Unknown	Maternal
	*SLC4A4*	NM_001098484.2:c.149G>C p.(Gly50Ala)	VUS	Het	AD/AR	Maternal
OFT-00851	*EML6*	NM_001039753.2:c.3181G>A p.(Asp1061Asn)	VUS	Het	Unknown	Maternal
	*COL9A3*	NM_001853.3:c.736C>T p.(Arg246Trp)	VUS	Het	AD	Paternal
	*TTN*	NM_133378.4:c.10361-3546T>C	VUS	Het	AR	Paternal
OFT-00852	*SLC4A11*	NM_001174090.1:c.2305G>A p.(Gly769Arg)	VUS	Het	AD	Unknown
	*LRP1B*	NM_018557.2:c.12983A>G p.(Tyr4328Cys)	VUS	Het	Unknown	Unknown
	*PIK3CG*	NM_002649.3:c.850G>A p.(Glu284Lys)	VUS	Het	Unknown	Unknown
	*COL6A6*	NM_001102608.1:c.5497T>C p.(Tyr1833His)	VUS	Het	Unknown	Unknown
	*LOXHD1*	NM_144612.6:c.2353G>A p.(Ala785Thr)	VUS	Het	AR	Unknown
OFT-00853	*LRP1B*	NM_018557.2:c.2568C>G p.(His856Gln)	VUS	Het	Unknown	Paternal
	*ALDH1A2*	NM_003888.3:c.173A>G p.(Tyr58Cys)	VUS	Het	Unknown	Maternal
	*CDHR1*	NM_033100.3:c.1397A>G p.(Asn466Ser)	VUS	Het	AR	Paternal
OFT-00854	*IPO5*	NM_002271.4:c.929C>T p.(Ala310Val)	VUS	Het	Unknown	Maternal
	*GALNT14*	NM_024572.3:c.472G>A p.(Asp158Asn)	VUS	Het	Unknown	Maternal
	*COL8A2*	NM_005202.3:c.1645G>A p.(Gly549Ser)	VUS	Het	AD	Maternal
OFT-00855	*ADAMTS9*	NM_182920.1:c.3967C>T p.(Arg1323Cys)	VUS	Het	Unknown	Unknown
	*ADAMTS8*	NM_007037.5:c.707G>A p.(Gly236Glu)	VUS	Het	Unknown	Unknown
OFT-00856	*EML6*	NM_001039753.2:c.2752A>G p.(Thr918Ala)	VUS	Het	Unknown	Paternal
	*DOCK9*	NM_001130048.1:c.5018G>A p.(Arg1673Gln)	VUS	Het	Unknown	Maternal
	*ADAMTS7*	NM_014272.4:c.646C>T p.(Arg216Trp)	VUS	Het	Unknown	Paternal
	*COL5A3*	NM_015719.3:c.2194G>A p.(Glu732Lys)	VUS	Het	Unknown	Paternal
OFT-00857	*ZNF469*	NM_001127464.2:c.1475G>A p.(Arg492Gln)	VUS	Het	AR	No Maternal
	*IDUA*	NM_000203.4:c.1354C>A p.(Arg452Ser)	VUS	Het	AR	Maternal
	*LOXL1*	NM_005576.3:c.1672C>G p.(His558Asp)	VUS	Het	Unknown	Maternal

^1^ VUS: variant of uncertain significance; ^2^ LP: probably pathogenic variant; ^3^ Het: heterozygosis; ^4^ AR: autosomal recessive; ^5^ AD: autosomal dominant. * ACMG results follow the American College of Medical Genetics and Genomics (ACMG) criteria included in Table A3 of Appendix A.

## Data Availability

Not applicable.

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
