# Peer review of "Whole-Exome Sequencing of 24 Spanish Families: Candidate Genes for Non-Syndromic Pediatric Keratoconus"

_genes, 2023, doi:10.3390/genes14101838_

Round 1
Reviewer 1 Report
This is a very thorough research and I believe it should be published. The authors have catalogued candidate genes for keratoconus from the whole exome sequencing of 24 families, based on what is known to be the pool of genes responsible for the structure, function, cell adhesion and development of the cornea. It should add a good deal of data to the available literature concerning the pathogeny and inheritance of this disease.
In the text there are small inadvertent affirmations, such as "spectacles [...] which only seems to be effective when they also present irregular astigmatism" -actually, spectacles usually do not correct irregular astigmatism.
Finally, I would be very interested to read in the future about the segregation studies that the authors are suggesting.
The English language is of good quality and well comprehensible.
There is alternate use of "frustrated" and "frustre" keratoconus. The usual expression is "forme fruste keratoconus (FFC)"
Fleck's Corneal Dystrophy is a misspell - correct is Fleck Corneal Dystrophy.
Author Response
Thank you very much for your time and comments. As you indicated, I believe they have been very useful to increase the quality and possible impact of the manuscript.
We would also like to thank you for your interest and your positive evaluation of the work. It has been an arduous task, but we believe that the results will be useful in the future for patients diagnosed with Keratoconus.
A Word document with the corrections one by one is attached.

Reviewer 2 Report
González-Atienza et al. realized a very interesting article describing the “Whole Exome Sequencing of 24 Spanish Families: Candidate Genes for Non-Syndromic Pediatric Keratoconus”. This manuscript examines the genetic basis of pediatric keratoconus in 24 Spanish families using whole exome sequencing. The topic is highly relevant given the lack of effective treatment options for this condition. The study design using trio/family-based exome sequencing is appropriate to identify potential candidate genes. The results identifying variants in genes related to corneal structure and function are intriguing. However, the manuscript would benefit from certain revisions to strengthen the presentation and interpretation of the findings.
Specific Comments:
- The introduction gives good background on keratoconus but could expand on current knowledge gaps and rationale for studying the genetic basis in pediatric cases specifically.
- In the methods, indicate whether the 24 families are unrelated and if any consanguinity was present.
- Provide more details on the variant filtering process and criteria for selecting possible disease-causing variants from the exome data.
- The results should start with a summary of the cohort characteristics like age, sex, family history patterns etc. before diving into the genetic findings.
- For the identified variants in each family, add details like zygosity, inheritance pattern, and whether the variants are novel or previously reported.
- In the discussion, put the identified variants into context of previously published keratoconus gene associations. Do your results support or contradict previous findings?
- Comment on whether you found an oligogenic inheritance pattern suggestive of multiple variant contributions.
- Discuss limitations like small sample size and lack of functional validation of the proposed candidate variants.
- The conclusion could be expanded to provide implications of your findings and suggest future research directions.
Overall, this is an interesting study that adds to the knowledge on the complex genetic basis of keratoconus. Addressing the above points would help improve the clarity and impact of the manuscript.
The English should benefit of an overall improvement.
Author Response

(The authors gave the same response as above.)
